# Volunteering and political participation are differentially associated with eudaimonic and social well-being across age groups and European countries

**Maria K. Pavlova**[ID]*, **Matthias Lühr**[¤]

Institute of Gerontology, University of Vechta, Vechta, Germany

¤ Current address: Work Group on Suicide Prevention, Section Public Mental Health, Psychiatric and Psychotherapeutic Clinic II, University of Ulm, Ulm, Germany
* maria.pavlova@uni-vechta.de

**Data Availability Statement:** The data underlying the results presented in the study are available at https://www.europeansocialsurvey.org/.

## Abstract

Voluntary participation is thought to promote the well-being of engaged individuals, especially in old age, but prior evidence on this link is mixed. In the present studies, we used the cross-sectional data from Round 6 (2012) of the European Social Survey (ESS) to investigate the variation in the associations between voluntary participation and eudaimonic (e.g., sense of direction) and social (e.g., perceived social support) well-being across types of participation (nonpolitical volunteering vs. political participation), age groups, and European countries. Study 1 addressed individual-level associations and age differences therein (preregistered at https://osf.io/2p9sz and https://osf.io/6twqe). Two-level multiple regression with an extensive set of control variables showed that at the within-country level, the associations between voluntary participation and well-being indicators were small on average. Nonpolitical volunteering had significantly more positive effects than did political participation, whereas few significant age differences emerged. Study 2 focused on the country-level variables that might explain the substantial cross-national variation in the main effects of voluntary participation (preregistered at https://osf.io/mq3dx). Only GDP per capita was a significant moderator at the country level: The associations of nonpolitical volunteering with eudaimonic well-being were more positive in the European countries with lower GDP. Other country-level variables (Gini coefficient, social welfare spending, and democracy indices) yielded no consistent moderation effects. Study 3 considered potential country-level explanations for the substantial cross-national variation in whether younger or older adults appeared to benefit more (preregistered at https://osf.io/7ks45). None of the country-level variables considered (effective retirement age in men, life expectancy at 65, average age of members of the national parliament and cabinet, and youth unemployment rate) could account for this variation. We conclude that, given the large cross-national variation in the effects of voluntary participation on well-being and in age differences therein, more attention to national specifics is warranted.

**Funding:** This work was supported by a German Research Foundation (DFG) grant (grant number PA 2704/3-1) to the first author. During the work on these studies, both the first and the second author were employed at the University of Vechta and received salaries from this public German university. There was no additional external funding received for this study.

**Competing interests:** The authors have declared that no competing interests exist.

## Introduction

Volunteering belongs to activities that promote well-being—it appears an established knowledge that it does, especially in old age [1–3]. As volunteering aims at contributing to the common good, it is hailed as a unique "win-win" situation for the engaged individuals and the society [4]. Although many studies delivered supporting evidence (see reviews in [1–3]), recent research with more robust designs (e.g., large-scale panel studies or randomized controlled trials, RCTs) brought rather disappointing news: The effects of volunteering on well-being turned out to be trivial, if at all significant, and inconsistent across age groups and countries [5–10].

In the present studies, which utilized the data from Round 6 (2012) of the European Social Survey (ESS), we adopted a differentiated view on the presumable benefits of volunteering. First, we asked: Which activity are we talking about? Prior research has usually implied nonpolitical formal volunteering, which is an organized voluntary activity to help others (or to achieve another common good, such as cleaning a river bank) directly [11, 12]. We juxtaposed this activity with conventional political participation: attempts to influence policy at different levels, which include not only political volunteering but also actions such as attending a demonstration [11]. Although political participation is likewise directed at the common good, namely policy change or social change, it may be more difficult and conflict prone than is nonpolitical volunteering [13, 14]. Both kinds of voluntary participation can be sometimes difficult to differentiate from informal volunteering, which is not attached to a formal organization but still involves unpaid voluntary work (e.g., providing help to nonkin [12]). However, the standard indices of volunteering and political participation that we used in the present studies referred explicitly to formal volunteering for nonpolitical or political purposes (i.e., "for organizations") or listed actions that cannot be considered as unpaid work (e.g., "contacted a politician" or "signed a petition") but qualify as political participation.

Our second question was, which outcomes? Subjective well-being (SWB, a hedonic aspect of well-being [15]) has been the most studied well-being outcome of volunteering [1], apart from its well-established effects on physical health [1–3]. However, it may not be the most proximal or enduring outcome. As Son and Wilson [16] argued, whereas the effects on SWB may be transient, in the long term, volunteering may provide opportunities for self-realization (eudaimonic well-being [17]) and improve the quantity and quality of social relationships (social well-being [18]). We therefore focused on eudaimonic and social well-being and considered their specific dimensions, which may yield differential effects of nonpolitical volunteering and political participation. For this reason, we used ESS data from 2012, the latest round when a comprehensive personal and social well-being module was administered.

Third, we asked: Who benefits most? Theoretically, the greatest benefits are expected in old age, when volunteering may compensate for role losses and keep one active [1–3]. In fact, few studies tested for age differences in the effects, and only a handful of those distinguished between nonpolitical and political activities or considered other well-being outcomes than SWB [9, 10, 19].

As this research yielded inconsistent findings across countries (e.g., Germany vs. the UK), our fourth and final question was, where are more benefits experienced, and why? Prior research has already addressed factors such as country-level volunteering rate, rate of social spending, and level of democracy as potential sources of variation in the link between volunteering or political activism and well-being, but findings remain inconclusive [20–27]. In the present article, we formulate and test specific expectations about which country-level variables moderate the associations between which activities (i.e., nonpolitical volunteering vs. political participation) with which outcomes (i.e., dimensions of eudaimonic and social-well-being) and in which age groups.

## Well-being outcomes of voluntary participation

The ESS well-being module is based on the multidimensional concept of flourishing, which includes aspects of hedonic ("feeling") and eudaimonic ("doing" or "functioning") well-being within the dimensions of personal and social well-being [28, 29]. In addition, it covers five ways to well-being: connect, be active, take notice, keep learning, and give [28]. In this broad conceptual framework, voluntary participation itself may be categorized as a "doing" aspect of social well-being (social engagement, caring, and altruism [29]) or as a way to well-being (connect, be active, and give [28]). In Table 1, we spell out conceptual reasons why nonpolitical volunteering and political participation may lead to improvement on the specific aspects of eudaimonic and social well-being that do not overlap with participation measures from the ESS module. Below, we review empirical evidence on the effects of voluntary participation on these well-being dimensions. It should be noted that many of these outcomes can be simultaneously considered as incentives or motivations to volunteer, which have been investigated in a separate line of research literature (including cross-national differences therein [30]). There is no contradiction, however, in that individuals volunteer in order to gain something, be it meaning of life or social connections, and that volunteering may indeed be instrumental in gaining that something. In the present paper, we focus on this latter aspect: documented outcomes of volunteering.

**Table 1. Theoretical links between nonpolitical volunteering and political participation and the outcome variables.**

| Outcomes | Nonpolitical volunteering (direct helping) | Political participation (changing policy) |
|---|---|---|
| **Eudaimonic well-being** | | |
| **Mastery (sense of competence)** | May improve mastery by making or achieving something of value to others | May improve mastery by achieving policy change (however, failed attempts may backfire) |
| **Learning experiences** | May provide learning opportunities in specific skills (e.g., cooking, cleaning, coaching) or general skills (e.g., organization, presentation, or speaking foreign language) and enhance knowledge of social issues | May provide learning opportunities in general skills (e.g., organization, presentation, or speaking foreign language) and enhance knowledge of social, economic, and political issues |
| **Purpose in life (sense of direction)** | May enhance purpose in life because others need one's help ("mattering") + sharing a goal with others | May enhance purpose in life because one perceives a need for social or policy change and sees ways to achieve it + sharing a goal with others (however, failed attempts may backfire) |
| **Engagement (flow)** | May fulfil the criteria for flow (right challenge + right skill), because volunteers are usually matched to tasks or can choose the tasks that provide optimal challenge | Although task matching and choosing tasks may apply here as well, attempts to change policy may overwhelm one's ability |
| **Social well-being** | | |
| **Generalized trust** | May enhance trust in strangers through positive, fulfilling contacts with other volunteers and help recipients | May enhance trust in strangers through collaboration with others to effect social change; however, political competition and negotiation may diminish trust |
| **Perceived social support** | May provide additional sources of social support by making friends with other volunteers | May provide additional sources of social support by making friends with other activists; however, political conflict may result in losing other sources of support (e.g., in the family) |
| **Low loneliness** | May decrease loneliness by connecting one to like-minded people | May decrease loneliness by connecting one to like-minded people; however, political conflict may result in feeling alone with one's political views (e.g., in the family) |

**Eudaimonic well-being.** Several studies used aggregate measures of psychological or eudaimonic well-being, which encompassed mastery or sense of control, sense of purpose in life, self-actualization, and related constructs, and found positive longitudinal effects of volunteering thereupon [16, 31] as well as positive associations with conventional political activism [32]. However, specific findings on mastery and purpose in life are mixed and mostly limited to nonpolitical volunteering in older adults [6, 7, 9, 33–35]. As regards political participation, Lühr et al. [9] found younger and middle-aged (but not older) Germans to report lower perceived control on occasions when they reported more political volunteering.

Two other eudaimonic well-being outcomes appear highly relevant to voluntary participation (see Table 1). In research on youth, civic engagement is understood as a natural learning context ("learning by doing"), which was shown to improve academic, personal, and social competence [36]. Service learning, programs that are designed to foster civic skills and volunteering in youth, does appear to promote understanding of social issues as well as organizational and academic skills [37]. In older adults, voluntary participation is presumed to provide opportunities for lifelong learning [38], an assumption supported by subjective reports of older activists and volunteers [39, 40].

Finally, engagement is part of the PERMA (positive emotion, engagement, relationships, meaning, and accomplishment) model of flourishing [41] and is closely related to the concept of flow [42]. It is characterized by high interest, concentration, and involvement in what one is doing [28, 29, 41]. Surprisingly little evidence is available on the link between voluntary participation and engagement [cf. 43], although the so-called serious leisure, which includes both career volunteering and casual volunteering, was shown to induce flow experience [44].

**Social well-being.** Generalized trust means trusting people in general; along with voluntary participation, it is seen as a key dimension of social capital [45, 46]. It has been argued that generalized trust and voluntary participation, which involves cooperating with (initial) strangers and helping strangers, are mutually reinforcing [45, 46]. However, international findings on the effects of voluntary participation on generalized trust are mixed [47–52].

Furthermore, perceived social support (i.e., beliefs that help, esteem, caring, and understanding are available when needed [53]) and loneliness (i.e., a subjective, aversive feeling that one's social relationships are qualitatively or quantitatively deficient [54]) refer to relationships with significant others. Again, international research that focused on nonpolitical volunteering reported mixed evidence on both outcomes [33, 55–60]. The effects of political participation have been understudied, but Lühr et al. [9, 10] found no or unfavorable effects of political volunteering on social support and loneliness.

## Age differences

A widespread opinion in the literature is that older adults benefit from volunteering more than other age groups do [1, 3, 61–64], because volunteering in older age may compensate for a loss of work and family roles, is truly voluntary and more intrinsically motivated (e.g., career considerations are no longer relevant), or involves more enjoyable and socially embedded activities (e.g., thanks to its more frequent religious underpinnings). Lühr et al. [9, 10] suggested differentiating between nonpolitical and political volunteering in this regard. Drawing on socioemotional selectivity theory [65], they argued that older adults, who have a shorter future time perspective and may therefore focus on more immediate goals and emotionally fulfilling relationships, may particularly benefit from nonpolitical volunteering, which typically involves harmonious social interactions and can yield comparatively quick results (e.g., achieving immediate relief for help recipients). By contrast, political participation implies competition with political opponents and sometimes even interpersonal conflict, whereas political

goals are more distant und difficult to achieve (i.e., changing policy [13, 14]). Hence, Lühr et al. [9, 10] argued that older adults might experience poorer outcomes of political participation.

Support for the notion that older volunteers experience greater benefits to SWB was found in two longitudinal studies from the USA [61, 64] and one cross-sectional [63] and one longitudinal [19] (here, very old adults benefited less than those around the retirement age) studies from Germany. Müller et al. additionally found stronger positive associations between volunteering and self-efficacy in older than in middle-aged adults. Moreover, Mannarini et al. [66] reported that older Italians who volunteered reported higher satisfaction with social relationships and leisure than their unengaged counterparts. In the most comprehensive test of age differences in the effects of voluntary participation on well-being to date, Lühr et al. [9, 10] found no significant age differences in Germany (outcomes included internal control beliefs, political efficacy, loneliness, social support availability, and SWB) and in the UK (outcomes included perceived social support, neighborhood belonging, and SWB). However, in German older adults, nonpolitical and political volunteering did show effects in the opposing directions (favorable for nonpolitical, unfavorable for political volunteering [9]), whereas this was not the case in their UK counterparts [10].

## Cross-national differences

Discrepant findings from different countries may result from random sampling errors or from true cross-national differences. It is well known that both the rates and the content of voluntary participation vary considerably across countries [12, 67, 68]. Several country-level characteristics that may be responsible for this huge variation have been proposed and tested [12, 67]. The most important of them include the level of democracy [69] or political rights and liberties [67], economic development (producing a sizable middle class, who are most likely to get engaged [69]), income inequality (presumably undermining interpersonal trust, exacerbating differences, magnifying political power of the affluent, and dampening collective efficacy of disadvantaged groups [67, 70]), state welfare spending ("crowding out" service activities of voluntary organizations while supporting participation of disadvantaged groups [68, 71]), institutional regime (influencing the size of the nonprofit sector and its dominant activities [68]), and religious tradition [67, 68] (higher religiosity but especially non-conservative religions, such as Protestantism, and multiconfessionalism promote voluntary participation [69]).

By contrast, cross-national comparative research on the outcomes of voluntary participation has been scarce and usually focused on SWB and self-reported physical or mental health as outcomes. For instance, Lee [72] found older adults' confidence in their ability to participate in politics to be associated with somewhat higher SWB in Western and Eastern and Central Europe but not in Nordic or Southern European countries. Rodríguez-Pose and von Berlepsch [73] reported that the small associations of volunteering with individual happiness in adults did not appear to differ across European regions, whereas indicators of political participation had significant (positive or negative) effects mainly in Western Europe. Using the Gallup World Poll data, Calvo et al. [74] and Kumar et al. [75] reported volunteering to have consistent associations with SWB and self-rated health across different world regions, even though volunteering yielded unfavorable associations with negative feelings only in low-income countries.

Other researchers considered country-level explanatory variables, such as volunteering rates and social spending. High volunteering rates may indicate a stronger social norm of and recognition for voluntary participation, whereas low rates may make volunteers more needed and their activities more rewarding [22]. Furthermore, "crowding out" the third sector

through high social spending may make volunteers feel less needed; alternatively, more voluntary participation of more diverse social groups, which is fostered by high social spending, may lead to more beneficial outcomes of participation [20]. Empirical findings on the roles of these country-level variables in the link between volunteering and SWB or self-rated health remain inconclusive [20, 22, 23, 25, 26]. However, using the ESS data, Plagnol and Huppert [27] found that formal volunteering was more strongly associated with higher SWB, sense of accomplishment, and feelings of doing something worthwhile in European countries with low (vs. high) volunteering rates.

Furthermore, several cross-national studies considered generalized trust as an outcome. Van Ingen and Bekkers [52] found volunteering to predict an increase in generalized trust over time only in the UK but not in Switzerland, the Netherlands, or Australia. Almakaeva et al. [21] used data from the World Values Survey and found political activism, but not participation in voluntary organizations, to be more strongly associated with higher generalized trust in countries with a higher human empowerment index (a combination of GDP per capita, emancipation values, and citizens' rights). In contrast, Kim [24] did not find the level of democracy to explain the variation in the link between political participation and generalized trust across Asian countries. We are not aware of any research that would address further outcomes or cross-national differences in age differences in the outcomes.

## Overview of the present studies

In the present studies, we investigated age and cross-national differences in the links between nonpolitical volunteering, political participation, and eudaimonic and social well-being. All studies utilized data from the ESS Round 6; their hypotheses, codebooks, and analytical methods were preregistered at the Open Science Framework. Study 1 focused on the individual-level associations between voluntary participation and well-being and on age differences therein. Study 2 addressed country-level factors that might explain cross-national variation in the main effects of voluntary participation. Study 3 considered country-level factors that might account for cross-national variation in age differences in the effects of voluntary participation.

## Study 1

The purpose of Study 1 was to test theory-driven hypotheses regarding age differences in the associations between voluntary participation and well-being (see preregistrations at https://osf. io/2p9sz and https://osf.io/6twqe). Generally, we expected both nonpolitical volunteering and political participation to have positive associations with eudaimonic (Hypothesis 1a) and social (Hypothesis 1b) well-being (see Table 1). Furthermore, we built on Lühr et al.'s [9, 10] ideas that nonpolitical volunteering is better suited to satisfy older adults' socioemotional needs than is political participation [cf. 65]. As additional factors, such as role losses and intrinsic motivation, may make nonpolitical volunteering especially rewarding in old age [1, 3, 61–64], we expected nonpolitical volunteering to have more positive associations with eudaimonic (Hypothesis 2a) and social (Hypothesis 2b) well-being in older adults in comparison to both younger and middle-aged adults. However, for learning experiences, we expected an exception to this pattern. Reporting learning experiences in everyday life is far less common in older age [28], which is unsurprising because older adults possess more life and work experience than younger adults do. As mentioned above, learning experiences are mainly seen as a typical outcome of youth volunteering and service learning [36, 37]. We therefore expected nonpolitical volunteering to have the less positive associations with learning experiences, the older the participants (Hypothesis 2c). Finally, as the factors that may reduce the rewards of political participation (i.e., a shortening time perspective, a changing goal focus) build up gradually over the

life span [65], we hypothesized that political participation would have the less positive associations with both eudaimonic (Hypothesis 3a) and social (Hypothesis 3b) well-being, the older the participants.

## Materials and methods

**Sample and procedure.** The ESS Round 6 [76] was a multi-national, cross-sectional survey conducted in 2012/2013 in 29 countries: Albania, Belgium, Bulgaria, Cyprus, Czech Republic, Denmark, Estonia, Finland, France, Germany, Hungary, Iceland, Ireland, Israel, Italy, Kosovo, Lithuania, Netherlands, Norway, Poland, Portugal, Russian Federation, Slovakia, Slovenia, Spain, Sweden, Switzerland, Ukraine, and United Kingdom. The survey targeted individuals over 14 years of age who resided in private households. All participating countries used some variant of probability sampling (e.g., simple, stratified, or multistage) to draw their samples. Response rates varied from 33.8% (Germany) to 78.1% (Israel), with an average rate of 61.8%. Data collection mode was a face-to-face interview, computer-assisted or paper-and-pencil, which lasted about one hour. Interviews were administered in local languages. The number of valid interviews varied from 752 (Iceland) to 2,958 (Germany); total $N = 54,673$.

**Measures.** Descriptive statistics for all study variables are shown in Table 2.

*Voluntary participation.* Frequency of nonpolitical volunteering was assessed with one item ("In the past 12 months, how often did you get involved in work for voluntary or charitable organisations?"; 1 = *never*; 6 = *at least once a week*). To ascertain the nonpolitical nature of this activity, we set the value on this variable at missing for 461 participants who scored above 1 on this item but indicated elsewhere in the questionnaire to have worked for a political party or action group but not for another voluntary organization. Political participation index was a mean on six items, all referring to the past 12 months: contacted a politician, government or local government official; worked in a political party or action group; worn or displayed a campaign badge/sticker; signed a petition; taken part in a lawful public demonstration; and boycotted certain products (0 = *no*; 1 = *yes*; α = .61).

*Eudaimonic well-being.* The factor structure of the personal and social well-being module from the ESS Round 6 was shown to vary substantially across participating countries [77]. We selected the following items and scales on the basis of their face validity and the evidence that the corresponding items often loaded on separate factors [77]. Three items assessed flow experiences: "How much of the time would you generally say you are. . . interested in/absorbed in/enthusiastic about what you are doing?" (0 = *none of the time*; 10 = *all of the time*; α = .91). One item assessed sense of direction: "To what extent do you feel that you have a sense of direction in your life?" (0 = *not at all*; 10 = *completely*). Four items assessed sense of competence (e.g., "Most days I feel a sense of accomplishment from what I do"; 1 = *disagree strongly*; 5 = *agree strongly*; α = .59). Finally, one item assessed learning new things: "Using this card, please tell me to what extent. . . you learn new things in your life?" (0 = *not at all*; 6 = *a great deal*).

*Social well-being.* Generalized trust was measured with three items ([78]; e.g., "Would you say that most people can be trusted, or that you can't be too careful in dealing with people?"; 0 = *you can't be too careful*; 10 = *most people can be trusted*; α = .78). Perceived social support was assessed with three items with different rating scales ($r$ = .31–.46): "To what extent do you feel appreciated by the people you are close to?" (0 = *not at all*; 10 = *completely*), "To what extent do you receive help and support from people you are close to when you need it?" (0 = *not at all*; 6 = *completely*), and "To what extent you feel that people treat you with respect?" (0 = *not at all*; 6 = *a great deal*). Finally, loneliness was measured with one item: "Please tell me how much of the time during the past week you felt lonely?" (1 = *none or almost none of the time*; 4 = *all or almost all of the time*; for previous use, see [79]).

**Table 2. Descriptive statistics for the central study variables at the individual level (Study 1).**

| Variable | Summary statistics [a] | | |
|---|---|---|---|
| | *M (SD)* | % | % of missing values |
| **Flow experiences [b] (0–10)** | 7.33 (1.85) | – | 0.8% |
| **Sense of competence [b] (1–5)** | 3.68 (0.59) | – | 0.9% |
| **Sense of direction (0–10)** | 7.00 (2.20) | – | 2.0% |
| **Learning new things (0–6)** | 4.06 (1.55) | – | 1.0% |
| **Generalized trust [b] (0–10)** | 5.09 (2.01) | – | 0.1% |
| **Perceived social support [b] (0–10)** | 5.75 (1.32) | – | 0.1% |
| **Loneliness (1–4)** | 1.47 (0.75) | – | 0.7% |
| **Frequency of nonpolitical participation (1–6)** | 1.95 (1.57) | – | 1.8% |
| **Index of political participation (0–1)** | 0.11 (0.18) | | 0.3% |
| **Age 15–30** | – | 21.1% | 0.2% |
| **Age 31–60** | – | 49.8% | 0.2% |
| **Age 61+** | – | 29.1% | 0.2% |
| **Female** | – | 54.4% | 0.0% |
| **Cohabiting with a partner** | – | 58.5% | 0.4% |
| **Children in household** | – | 38.4% | 0.0% |
| **In education or training** | – | 8.4% | 1.8% |
| **Unemployed** | – | 5.8% | 1.8% |
| **Out of the labor market** | – | 35.5% | 1.8% |
| **Own educational attainment (1–7)** | 3.87 (1.85) | – | 0.7% |
| **Parents' educational attainment (1–7)** | 3.15 (1.97) | – | 4.6% |
| **Partner's educational attainment (1–7)** | 3.98 (1.84) | – | 42.9% |
| **Work-related further education in the past 12 months** | – | 28.0% | 0.8% |
| **Occupational autonomy (0–10)** | 5.14 (3.27) | – | 11.2% |
| **Net household income decile (1–10)** | 5.06 (2.81) | – | 19.6% |
| **General health (1–5)** | 3.75 (0.94) | – | 0.2% |
| **Physical activity (0–7)** | 4.75 (2.45) | – | 1.5% |
| **Socializing with others (1–7)** | 4.79 (1.63) | – | 0.6% |
| **Attendance at religious services (1–7)** | 2.59 (1.53) | – | 0.8% |
| **Benevolence[c] (-3.00–3.81)** | 0.64 (0.66) | – | 1.0% |
| **Universalism[c] (-3.57–3.24)** | 0.51 (0.63) | – | 0.8% |
| **Achievement[c] (-4.13–3.25)** | -0.31 (0.90) | – | 1.1% |
| **Stimulation[c] (-4.60–3.94)** | -0.71 (1.00) | – | 1.1% |
| **Hedonism[c] (-4.56–3.19)** | -0.32 (0.99) | – | 1.1% |
| **Power[c] (-4.42–3.71)** | -0.80 (0.92) | – | 0.9% |

*Note*. Dash = not applicable.

[a] Summary statistics across all individuals.

[b] Mean score of the indicators of flow experiences, sense of competence, generalized trust, and perceived social support, respectively.

[c] Each value is represented by a mean score of three person-mean centered items.

*Moderator variable*. As our hypotheses sometimes assumed nonlinear moderating effects of age, we divided participants into three age groups based on their chronological age: younger adults (15–30), middle-aged adults (31–60), and older adults (60+).

*Control variables*. Prior research identified a lot of third variables that may influence both voluntary participation and well-being [9–12] and therefore need to be considered as potential

confounders in a correlational study [80]. Sociodemographic and socioeconomic indicators included sex (0 = *male*; 1 = *female*), partnership status (0 = *not cohabiting*; 1 = *cohabiting with a partner*) and having children under 16 in the household (0 = *no*; 1 = *yes*), employment status (*in education or training*, *unemployed*, *out of the labor market*, or *employed*), one's own, partner's, and parents' highest educational attainment (ISCED classification: 1 = *less than lower secondary*; 7 = *upper tertiary*), work-related further education or training in the past 12 months (0 = *no*; 1 = *yes*), occupational autonomy in the current or previous job (a mean score of two items, e.g., "How much the management at your work allows/allowed you to decide how your own daily work is/was organised"; 0 = *no influence*; 10 = *complete influence*), and country-specific net household income decile (1–10). Health and leisure activities covered self-reported general health (1 = *very bad*; 5 = *very good*), frequency of physical activity in the past seven days (0 = *on no days*; 7 = *on all days*), frequency of socializing with friends, relatives, or work colleagues (1 = *never*; 7 = *every day*), and frequency of attendance at religious services (1 = *never*; 7 = *every day*). Finally, to take dispositional factors into account, we used six two-item scales from the Portrait Values Questionnaire (PVQ) [81] to assess the values of benevolence, universalism, achievement, stimulation, hedonism, and power, which are known to be associated with nonpolitical and political participation [82, 83] as well as with eudaimonic and social well-being [84, 85]. Before computing mean scores on each scale, we centered each item on the mean value across all 21 PVQ items [81].

**Analytical approach.** We conducted all analyses in Mplus v. 8.6 for Linux [86]. We modeled multiple-indicator dependent variables (i.e., flow experiences, sense of competence, generalized trust, and perceived social support) as latent variables at both individual and country levels (i.e., two-level CFA) and assessed measurement invariance across 29 countries using ML estimation [86, 87]. Subsequently, we conducted two-level multiple regression models with participants nested within countries [88]. Here, we used Bayesian estimation, which is especially well suited for multilevel analyses with a small number of units at the highest level [89] and for large computational problems [90]. However, because of lacking or inconsistent

**Table 3. Results from ML estimation of measurement models.**

| Model | $\chi^2$ (df) | CFI | RMSEA | SRMR$_{within}$ | SRMR$_{between}$ | Sample-size adjusted BIC | $\Delta\chi^2$ (df) |
|---|---|---|---|---|---|---|---|
| | | | *Flow experiences* (N = 54,251) | | | | |
| **Configural invariance** | 0.0 (0) | 1.000 | 0.000 | 0.000 | 0.000 | 575238.0 | n/a |
| **Weak invariance** | 0.6 (2), *ns* | 1.000 | 0.000 | 0.000 | 0.042 | 575223.2 | 0.6 (2), *ns* |
| **Strong invariance** | 2646.0 (5) *** | 0.974 | 0.099 | 0.005 | 0.112 | 577845.4 | 2645.4 (3) *** |
| | | | *Sense of competence* (N = 54,576) | | | | |
| **Configural invariance** | 235.8 (4) *** | 0.991 | 0.033 | 0.016 | 0.071 | 513753.8 | n/a |
| **Weak invariance** | 243.4 (7) *** | 0.991 | 0.025 | 0.016 | 0.159 | 513738.2 | 7.6 (3), *ns* |
| **Strong invariance** | 4981.0 (11) *** | 0.805 | 0.091 | 0.026 | 0.442 | 518444.9 | 4737.6 (4) *** |
| | | | *Generalized trust* (N = 54,607) | | | | |
| **Configural invariance** | 0.0 (0) | 1.000 | 0.000 | 0.000 | 0.000 | 689999.0 | n/a |
| **Weak invariance** | 2.0 (2), *ns* | 1.000 | 0.001 | 0.000 | 0.043 | 689985.5 | 2.0 (2), *ns* |
| **Strong invariance** | 2080.1 (5) *** | 0.943 | 0.087 | 0.010 | 0.060 | 692040.4 | 2078.1 (3) *** |
| | | | *Perceived social support* (N = 54,618) | | | | |
| **Configural invariance** | 0.0 (0) | 1.000 | 0.000 | 0.000 | 0.000 | 536895.4 | n/a |
| **Weak invariance** | 1.8 (2), *ns* | 1.000 | 0.000 | 0.000 | 0.081 | 536881.8 | 1.8 (2), *ns* |
| **Strong invariance** | 3891.5 (5) *** | 0.834 | 0.119 | 0.017 | 0.378 | 540748.3 | 3889.7 (3) *** |

*Note.* $\Delta\chi^2$ refers to the difference from the previous model (i.e., weak vs. configural invariance or strong vs. weak invariance). n/a = not applicable.

*** $p < .001$.

information on the country-level parameters from prior research, we employed noninformative priors. Moreover, we could not use weighting to correct for unequal selection probabilities as it is not available with Bayesian estimation [86]. (We conducted supplementary analyses of the random-intercept models described below with ML estimation using post-stratification weights and ascertained that the estimates from these analyses were very close to those reported in this article.) Missing values on both dependent and independent variables at the individual level were estimated directly in each model under MAR assumption: Bayesian estimator is a full information estimator, which uses all available information from each case and implies probable values in the place of missing values (assuming missing values are a function of available covariates) during the estimation of model parameters without inflating the effective sample size [90, 91].

First, we estimated random-intercept models (i.e., the intercept of the dependent variable was free to vary across countries [88]), which were the main focus of Study 1 as we were interested in individual-level associations. In unadjusted models (Models 1 in Tables 4 and 5), we entered only nonpolitical volunteering and political participation (cluster-mean centered), age groups (dummy coded with older adults as a reference category; uncentered, as the variance at the between level was negligible), and the interactions between age groups and each type of voluntary participation as predictors in the regression equation at the individual level. For interactions, we used an adjusted alpha level of .01 to avoid spuriously significant associations caused by multiple significance testing. In fully adjusted models (Models 2 in Tables 4 and 5), we added all individual-level control variables. We applied latent centering to dependent variables (i.e., modeling their means and variances at the country level as well) and observed cluster-mean centering to continuous covariates [86]. Second, we estimated random-intercept-and-slope models, where the slopes of nonpolitical volunteering or political participation, age groups, and their interactions were free to vary across countries [88]. This was a preparatory step for subsequent studies, because we needed to ensure that there was significant variation in the associations at the country level.

## Results

**Measurement model.** Table 3 shows fit indices for the measurement models and model comparison for configural (i.e., factor indicators are the same across levels), weak (i.e., factor loadings are constrained to be equal across levels), and strong (i.e., additionally, residual variances of factor indicators are fixed at zero at the between level) invariance assumptions [87]. Weak measurement invariance across countries was supported for all four latent variables; thus, the meaning of the latent constructs was the same at the individual and the country levels. Strong measurement invariance was not supported; thus, the means of the latent constructs were incomparable across countries, but regression coefficients would be comparable [92]. The fit of the models assuming weak measurement invariance ranged from acceptable to excellent.

**Multiple regression results.** *Eudaimonic well-being*. Tables 4 and 5 show findings for eudaimonic and social well-being outcomes, respectively. In unadjusted models (Model 1), all predictors had highly significant associations with eudaimonic outcomes. With interaction effects in the model, the main effects of nonpolitical volunteering and political participation referred to the effects among older adults (the reference group). These effects were positive and significant (see Table 4). In turn, almost all interactions with age groups were negative and significant (at $p < .01$), indicating that the effects of both nonpolitical volunteering and political participation were less positive in younger and middle-aged adults. This pattern of associations was not retained in fully adjusted models (Model 2). Effect magnitude was substantially

**Table 4. Individual-level effects of nonpolitical volunteering and political participation on eudaimonic well-being.**

| Predictors | Flow experiences | | Sense of direction | | Sense of competence | | Learning new things | |
|---|---|---|---|---|---|---|---|---|
| | Model 1 | Model 2 | Model 1 | Model 2 | Model 1 | Model 2 | Model 1 | Model 2 |
| **Nonpolitical volunteering** | 0.09** (0.01) | 0.05** (0.01) | 0.16** (0.01) | 0.05** (0.01) | 0.12** (0.01) | 0.06** (0.01) | 0.14** (0.01) | 0.08** (0.01) |
| **Political participation** | 0.44** (0.06) | 0.11 (0.06) | 0.75** (0.12) | 0.10 (0.11) | 0.52** (0.06) | 0.10 (0.07) | 1.13** (0.08) | 0.43** (0.07) |
| **Age 15–30[a]** | 0.13** (0.01) | -0.13** (0.02) | 0.28** (0.03) | -0.30** (0.04) | 0.18** (0.02) | -0.27** (0.02) | 1.28** (0.02) | 0.46** (0.02) |
| **Age 31–60[a]** | 0.10** (0.01) | -0.10** (0.02) | 0.21** (0.02) | -0.27** (0.03) | 0.24** (0.01) | -0.19** (0.02) | 0.77** (0.01) | 0.23** (0.02) |
| **Nonpolitical volunteering x Age 15–30** | -0.04** (0.01) | -0.02* (0.01) | -0.06** (0.02) | -0.03 (0.02) | -0.04** (0.01) | -0.02 (0.01) | -0.05** (0.01) | -0.04** (0.01) |
| **Nonpolitical volunteering x Age 31–60** | -0.03** (0.01) | -0.02* (0.01) | -0.04** (0.02) | -0.02 (0.01) | -0.03** (0.01) | -0.02 (0.01) | -0.06** (0.01) | -0.04** (0.01) |
| **Political participation x Age 15–30** | -0.28** (0.08) | -0.12 (0.08) | -0.71** (0.17) | -0.31 (0.16) | -0.34** (0.10) | -0.09 (0.10) | -0.73** (0.11) | -0.41** (0.11) |
| **Political participation x Age 31–60** | -0.15* (0.07) | -0.03 (0.07) | -0.22 (0.14) | 0.03 (0.13) | -0.18* (0.08) | 0.00 (0.08) | -0.40** (0.09) | -0.21** (0.08) |
| *Control variables* | | | | | | | | |
| **Female** | | -0.02 (0.01) | | 0.00 (0.02) | | 0.04** (0.01) | | 0.02 (0.01) |
| **Cohabiting with a partner** | | 0.14** (0.02) | | 0.30** (0.03) | | 0.21** (0.02) | | 0.10** (0.02) |
| **Children in the household** | | -0.04** (0.01) | | -0.04* (0.02) | | 0.05** (0.01) | | 0.01 (0.01) |
| **In education/training[b]** | | 0.13** (0.02) | | 0.23** (0.04) | | -0.04 (0.03) | | 0.40** (0.03) |
| **Unemployed[b]** | | -0.15** (0.02) | | -0.53** (0.04) | | -0.41** (0.03) | | -0.11** (0.03) |
| **Out of the labor market[b]** | | -0.03 (0.02) | | -0.12** (0.03) | | -0.21** (0.02) | | -0.21** (0.02) |
| **Own educational attainment** | | 0.03** (0.00) | | 0.06** (0.01) | | 0.04** (0.00) | | 0.07** (0.00) |
| **Parents' educational attainment** | | -0.01 (0.00) | | 0.00 (0.01) | | -0.01 (0.01) | | 0.00 (0.01) |
| **Partner's educational attainment** | | 0.00 (0.00) | | -0.01 (0.01) | | -0.01** (0.00) | | 0.03** (0.00) |
| **Further education/training past 12 months** | | 0.04** (0.01) | | 0.07** (0.02) | | 0.07** (0.02) | | 0.19** (0.02) |
| **Occupational autonomy** | | 0.04** (0.00) | | 0.07** (0.00) | | 0.05** (0.00) | | 0.04** (0.00) |
| **Net household income decile** | | 0.02** (0.00) | | 0.06** (0.01) | | 0.03** (0.00) | | 0.02** (0.00) |
| **General health** | | 0.21** (0.01) | | 0.43** (0.01) | | 0.30** (0.01) | | 0.23** (0.01) |
| **Physical activity past 7 days** | | 0.07** (0.00) | | 0.08** (0.00) | | 0.07** (0.00) | | 0.05** (0.00) |
| **Frequency of socializing** | | 0.05** (0.00) | | 0.11** (0.01) | | 0.07** (0.00) | | 0.06** (0.00) |
| **Attendance at religious services** | | 0.02** (0.00) | | 0.10** (0.01) | | 0.05** (0.00) | | 0.03** (0.00) |
| **Benevolence** | | 0.04** (0.01) | | 0.05** (0.02) | | 0.08** (0.01) | | 0.02 (0.01) |
| **Universalism** | | -0.02* (0.01) | | -0.09** (0.02) | | -0.07** (0.01) | | 0.07** (0.01) |
| **Achievement** | | 0.09** (0.01) | | 0.11** (0.01) | | 0.15** (0.01) | | 0.13** (0.01) |
| **Stimulation** | | 0.00 (0.01) | | 0.00 (0.01) | | 0.02** (0.01) | | 0.13** (0.01) |
| **Hedonism** | | -0.03** (0.01) | | -0.02* (0.01) | | 0.01 (0.01) | | 0.02** (0.01) |
| **Power** | | -0.15** (0.01) | | -0.15** (0.01) | | -0.13** (0.01) | | -0.08** (0.01) |

*(Continued)*

**Table 4.** (*Continued*)

| Predictors | Flow experiences | | Sense of direction | | Sense of competence | | Learning new things | |
|---|---|---|---|---|---|---|---|---|
| | Model 1 | Model 2 | Model 1 | Model 2 | Model 1 | Model 2 | Model 1 | Model 2 |
| $R^2_{within}$ | .020 | .150 | .015 | .130 | .039 | .253 | .123 | .235 |

*Note.* $N$ = 54,673. Cells represent unstandardized linear regression coefficients with standard errors in parentheses. Flow experiences and sense of competence were modeled as latent variables.

[a] Reference category: Age 61+.

[b] Reference category: Employed.

[*] $p < .05$

[**] $p < .01$.

reduced, whereby the effects of political participation and all interaction effects lost significance (at $p < .01$) for three out of four outcomes. We re-ran the analyses for these outcomes omitting the interaction effects to obtain more precise estimates of the main effects. In these models, nonpolitical volunteering was significantly associated with higher flow experiences, $B$ ($SE$) = 0.03 (0.00), $p < .01$, $\beta$ = .05, sense of direction, $B$ ($SE$) = 0.04 (0.01), $p < .01$, $\beta$ = .03, and sense of competence, $B$ ($SE$) = 0.05 (0.00), $p < .01$, $\beta$ = .06, whereas political participation was significantly associated with higher flow experiences, $B$ ($SE$) = 0.06 (0.03), $p < .05$, $\beta$ = .01, and sense of competence, $B$ ($SE$) = 0.07 (0.04), $p < .05$, $\beta$ = .01, and not associated with sense of direction, $B$ ($SE$) = 0.05 (0.06), *ns*. We also conducted post-hoc comparisons of the standardized main effects of nonpolitical volunteering and political participation on all eudaimonic well-being outcomes; the former were always significantly ($p < .01$) more positive than the latter (see Fig 1).

In adjusted models, significant age differences emerged only for learning new things. Fig 2A shows expected values of this outcome in different age groups depending on the amount of nonpolitical volunteering and political participation. To facilitate interpretation of effect sizes, the Y axis is scaled to one *SD* of the dependent variable at the within-country level. There was a strong main effect of age: the older the participants, the fewer learning experiences they reported. In contrast, the associations of nonpolitical volunteering and political participation with learning new things grew more positive with the participants' age, even though significant associations were observed in younger and middle-aged adults as well (with the exception of political participation in younger adults).

To sum up, Hypothesis 1a (generally positive effects of both nonpolitical volunteering and political participation) was supported for flow experiences and sense of competence but qualified by very small effect sizes. Additionally, across age groups, nonpolitical volunteering was significantly positively associated with sense of direction and learning new things. As regards age differences, Hypothesis 2a (the associations between nonpolitical volunteering and most eudaimonic outcomes are more positive in older adults) was not supported. We found this pattern of age differences only for learning new things, where we expected the opposite (Hypothesis 2c). Finally, Hypothesis 3a was not supported either, because age differences in the effects of political participation emerged only for learning new things and were likewise in unexpected direction.

*Social well-being.* Unadjusted models yielded generally favorable associations of both nonpolitical volunteering and political participation with almost all outcomes in older adults and significant age differences ($p < .01$) for loneliness (see Table 5, Models 1). However, these findings were not supported by fully adjusted models (see Table 5, Models 2). Again, where no significant interaction effects were found, we re-ran the analyses excluding the interactions.

**Table 5. Individual-level effects of nonpolitical volunteering and political participation on social well-being.**

| Predictors | Generalized trust | | Perceived social support | | Loneliness | |
|---|---|---|---|---|---|---|
| | Model 1 | Model 2 | Model 1 | Model 2 | Model 1 | Model 2 |
| Nonpolitical volunteering | 0.04** (0.01) | 0.01 (0.01) | 0.04** (0.01) | -0.01 (0.01) | -0.03** (0.00) | -0.01 (0.00) |
| Political participation | 0.35** (0.06) | 0.02 (0.06) | 0.01 (0.06) | -0.34** (0.07) | -0.17** (0.04) | 0.04 (0.04) |
| Age 15–30[a] | 0.02 (0.01) | -0.23** (0.02) | -0.01 (0.02) | -0.24** (0.02) | -0.19** (0.01) | -0.01 (0.01) |
| Age 31–60[a] | -0.05** (0.01) | -0.19** (0.02) | -0.11** (0.01) | -0.30** (0.02) | -0.16** (0.01) | 0.06** (0.01) |
| Nonpolitical volunteering x Age 15–30 | 0.01 (0.01) | 0.02* (0.01) | 0.02 (0.01) | 0.04** (0.01) | 0.02** (0.01) | 0.01 (0.01) |
| Nonpolitical volunteering x Age 31–60 | 0.01 (0.01) | 0.02* (0.01) | 0.00 (0.01) | 0.02 (0.01) | 0.01* (0.01) | 0.01 (0.01) |
| Political participation x Age 15–30 | -0.22* (0.09) | -0.11 (0.09) | 0.09 (0.10) | 0.22* (0.10) | 0.23** (0.06) | 0.09 (0.05) |
| Political participation x Age 31–60 | -0.06 (0.07) | -0.01 (0.07) | 0.20* (0.08) | 0.30** (0.08) | 0.16** (0.05) | 0.05 (0.04) |
| *Control variables* | | | | | | |
| Female | | 0.06** (0.01) | | 0.11** (0.01) | | 0.05** (0.01) |
| Cohabiting with a partner | | -0.04 (0.02) | | 0.14** (0.02) | | -0.46** (0.01) |
| Children in the household | | -0.04** (0.01) | | -0.02 (0.01) | | -0.01 (0.01) |
| In education/training[b] | | 0.20** (0.02) | | 0.18** (0.03) | | -0.14** (0.01) |
| Unemployed[b] | | -0.07** (0.02) | | -0.12** (0.03) | | 0.05** (0.01) |
| Out of the labor market[b] | | 0.05** (0.02) | | 0.04* (0.02) | | 0.07** (0.01) |
| Own educational attainment | | 0.04** (0.00) | | 0.02** (0.00) | | -0.01** (0.00) |
| Parents' educational attainment | | 0.02** (0.00) | | 0.01* (0.01) | | 0.02** (0.00) |
| Partner's educational attainment | | 0.01 (0.00) | | 0.01* (0.00) | | 0.00 (0.00) |
| Further education/training past 12 months | | 0.03* (0.01) | | 0.04** (0.01) | | -0.01 (0.01) |
| Occupational autonomy | | 0.02** (0.00) | | 0.03** (0.00) | | -0.01** (0.00) |
| Net household income decile | | 0.03** (0.00) | | 0.04** (0.00) | | -0.02** (0.00) |
| General health | | 0.15** (0.01) | | 0.21** (0.01) | | -0.14** (0.00) |
| Physical activity past 7 days | | 0.00 (0.00) | | 0.04** (0.00) | | -0.02** (0.00) |
| Frequency of socializing | | 0.04** (0.00) | | 0.12** (0.00) | | -0.04** (0.00) |
| Attendance at religious services | | 0.03** (0.00) | | 0.03** (0.00) | | 0.00 (0.00) |
| Benevolence | | 0.10** (0.01) | | 0.23** (0.01) | | -0.03** (0.01) |
| Universalism | | 0.15** (0.01) | | 0.00 (0.01) | | 0.00 (0.01) |
| Achievement | | 0.01 (0.01) | | 0.05** (0.01) | | 0.01** (0.00) |
| Stimulation | | 0.03** (0.01) | | -0.07** (0.01) | | 0.01** (0.00) |
| Hedonism | | 0.05** (0.01) | | 0.02* (0.01) | | -0.03** (0.00) |
| Power | | 0.02* (0.01) | | -0.11** (0.01) | | 0.03** (0.00) |
| $R^2_{within}$ | .010 | .078 | .010 | .173 | .015 | .161 |

*Note.* $N$ = 54,673. Cells represent unstandardized linear regression coefficients with standard errors in parentheses. Generalized trust and perceived social support were modeled as latent variables.

[a] Reference category: Age 61+.

[b] Reference category: Employed.

* $p < .05$

** $p < .01$.

Nonpolitical volunteering was significantly associated with higher generalized trust, $B$ ($SE$) = 0.02 (0.00), $p < .01$, $\beta$ = .03, but not with loneliness, $B$ ($SE$) = 0.00 (0.00), *ns*. In contrast, political volunteering was not associated with generalized trust, $B$ ($SE$) = 0.00 (0.03), *ns*, but it had a significantly positive (i.e., unfavorable) association with loneliness, $B$ ($SE$) = 0.09 (0.02), $p < .01$, $\beta$ = .02. A post-hoc comparison of all standardized main effects revealed again that the associations of nonpolitical volunteering with social well-being outcomes were significantly ($p < .01$) more positive than those of political participation (see Fig 1).

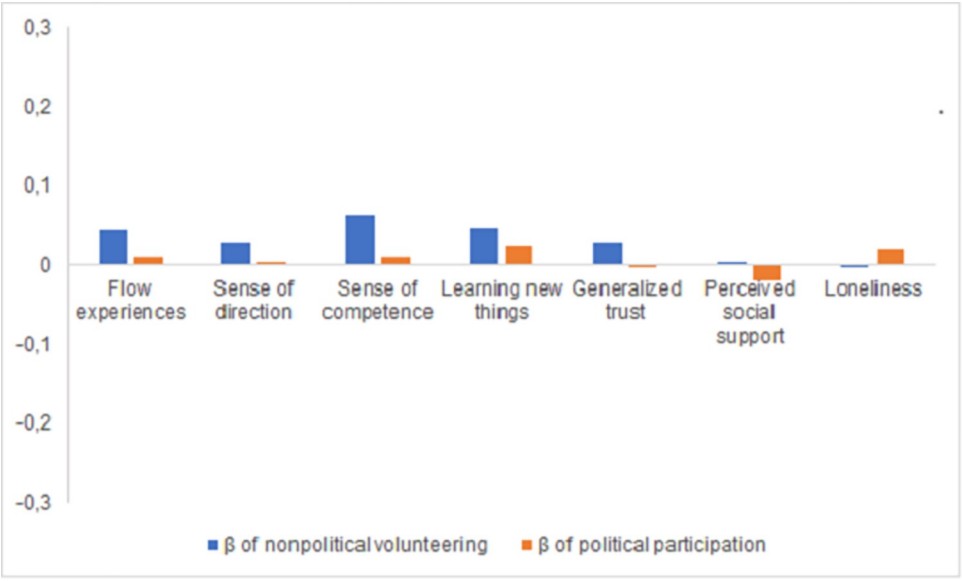

**Fig 1. Standardized effects of nonpolitical volunteering and political participation on well-being outcomes in comparison.** Within each pair, effects significantly differ at $p < .01$.

Age differences in the associations with perceived social support became significant ($p < .01$) in fully adjusted models. As shown in Fig 2b, there was a main effect of age: Older adults reported significantly higher social support than younger and middle-aged adults did. However, nonpolitical volunteering had a significantly positive association with perceived social support in younger adults only, whereas political participation had a significantly negative association with this outcome in older adults only.

Thus, Hypothesis 1b was not supported, as the only positive effect on social well-being that emerged "across the board" was the small association of nonpolitical volunteering with generalized trust. Hypothesis 2b was not supported, as nonpolitical volunteering did not have significantly more positive effects in older adults. By contrast, Hypothesis 3b was supported for perceived social support (but not for other outcomes), because the effect of political participation in older adults was more negative.

*Effects of control variables.* The following factors had consistently significant positive associations with eudaimonic well-being: partnership, being employed or in education, socioeconomic status, self-reported general health, all leisure activities considered, and the personal

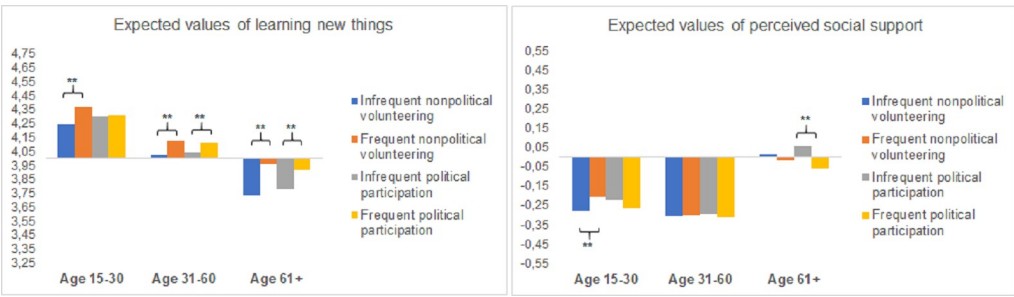

**Fig 2.** a. Effects of voluntary participation on learning new things are moderated by age. b. Effects of voluntary participation on perceived social support are moderated by age.

values of benevolence and achievement. Other personal values showed inconsistent or negative (power) associations with eudaimonic outcomes. For social well-being, consistently favorable associations emerged with female gender, being employed or in education, socioeconomic status, self-reported general health, frequency of socializing, and the values of benevolence and hedonism. Other personal values had inconsistent associations with social well-being.

By entering control variables in blocks (not shown in tables), we found out which of them were mainly responsible for the drastic difference in the effects of focal predictors between unadjusted and adjusted models. For both nonpolitical volunteering and political participation, these were health and leisure activities, and for political participation, these were additionally sociodemographics and SES. In contrast, entering personal values did not change the effects of focal predictors substantially.

*Cross-national variation in effects.* Allowing the slopes of the main effects and the interactions with age to vary at the between-country level showed that each slope had a significant variance at this level (see S1 Table).

## Discussion

In Study 1, we found small associations of nonpolitical volunteering and political participation with indicators of eudaimonic and social well-being in 29 European countries (ESS Round 6). Generally, nonpolitical volunteering had significantly more positive effects than did political participation (see Fig 1). This finding supports the view on political participation as a more cumbersome and less socially rewarding undertaking as compared to nonpolitical volunteering [13, 14] (see Table 1). However, given the cross-sectional nature of effects, it cannot be ruled out that even this small difference was due to a self-selection of individuals with higher well-being into nonpolitical volunteering rather than into political participation.

Contrary to our expectations, we found few age differences in the associations between voluntary participation and well-being. Exactly where we expected older adults to benefit less (i.e., in terms of learning experiences), significantly stronger positive associations in older adults emerged for both predictors. Voluntary participation may gain importance as a source of learning in old age, because its more conventional sources are lacking (i.e., formal education or training and a workplace). For perceived social support, findings hinted at older adults' benefiting less from both types of voluntary participation. First, nonpolitical volunteering unexpectedly had positive and significant associations with this outcome in younger but not in middle-aged or older adults. Second, political participation had significantly *negative* associations with perceived social support in older but not in younger or middle-aged adults. This result was in line with our expectation that older adults might be especially sensitive to interpersonal and intergroup conflict inherent to political participation [13, 14, 65], but it was limited to this particular outcome. Important to note, older adults reported to perceive considerably higher social support than both younger and middle-aged adults did, so the less favorable effects of voluntary participation did not imply any serious disadvantage of older adults.

We used an extensive set of control variables, including not only sociodemographic and socioeconomic indicators and personal values but also general health and other leisure activities, such as physical activity, informal socializing, and church attendance. The difference between unadjusted and adjusted models indicated that much of the association between voluntary participation and well-being might be attributed to self-selection, which may be especially potent in old age. For some of our control variables, it can be argued that they may mediate the associations between voluntary participation and well-being and should be treated as mediators rather than confounders [1, 2]. However, it would be disappointing if the benefits

of voluntary participation were mainly conferred through mere socializing or physical activity, both of which can be achieved in many ways without contributing to a common good. Similarly, although voluntary participation may also improve physical health [1, 2], theoretically, it should yield effects on outcomes such as eudaimonic well-being largely irrespective of health status (see Table 1). As health is a well-known prerequisite to voluntary participation [11, 12], in a cross-sectional study, it is more appropriate to treat it as a confounder [80].

Our findings at the individual level concurred with other recent research that failed to find substantial benefits of volunteering to well-being across the board [5–10]. However, we also found that there was a significant variation across European countries in the main effects of nonpolitical volunteering and political participation on each outcome considered. Likewise, the slopes of interactions with age varied significantly across countries. Thus, it was warranted to address potential explanations for this variation.

## Study 2

The purpose of Study 2 was to identify the country-level explanatory variables for the cross-national variation in the main effects of voluntary participation on eudaimonic and social well-being (see a preregistration at https://osf.io/mq3dx). As this endeavor required testing several cross-level interactions for each outcome (see Analytical Approach), we included only the most plausible outcomes in the hypotheses formulated below. Sticking to our central distinction between nonpolitical volunteering and political participation (see Table 1), we propose nonpolitical volunteering to be more beneficial in countries where the need (i.e., of help recipients) is greater, whereas political participation may yield more benefits in countries where it has greater chances of success.

Our reasoning is as follows. Nonpolitical volunteering aims to help others or to combat some other problem directly, typically on a small scale (i.e., community). Its aims are therefore easier to accomplish (in comparison to policy change) and are achievable even in countries with barriers to democratic participation and low economic development. Moreover, as helping others only makes sense when others need help, countries with poor economic conditions, high inequality, and low state welfare spending [67, 68, 71] may paradoxically feature low volunteering rates but comparatively high well-being benefits experienced by volunteers [22, 27]. In particular, in such countries, nonpolitical volunteering may feel more fulfilling (operationalized by flow experiences) and meaningful (sense of direction) and bring greater social recognition (perceived social support). We therefore expected the associations of nonpolitical volunteering with flow experiences, sense of direction, and perceived social support to be more positive in countries with lower economic development (Hypothesis 1a), higher income inequality (Hypothesis 1b), and lower state welfare spending (Hypothesis 1c). These country-level factors have all featured as potential determinants of volunteering rates [67, 68, 71], but only state welfare spending was used as a moderator of the link between volunteering and outcomes [20, 22].

Political participation pursues aims that are typically more difficult to achieve (i.e., policy change or social change) and typically involves more conflict and contestation [13, 14] (see Table 1). Both the success of political endeavors and the extent of conflict involved may crucially depend on the power relations in the society and their changeability. In particular, economic inequality kept in check, availability of free elections, and access of average citizens as well as minorities to democratic participation and political decision-making [21, 67, 69, 70] may all contribute to political participation taking more respectful and safe forms and bringing about the desired change (or achieving tolerable compromises between conflicting interests). We assumed that better prospects of success would lead to political participation being more

fulfilling (in terms of flow experiences) and boosting one's self-worth (sense of competence), whereas a lower conflict potential would foster more positive social well-being outcomes as regards relationships both with people in general (i.e., generalized trust) and with significant others (i.e., perceived social support and lower loneliness). Hence, we hypothesized that the associations of political participation with flow experiences, sense of competence, generalized trust, perceived social support, and lower loneliness would be more positive in countries with lower income inequality (Hypothesis 2a), a better access to free elections (Hypothesis 2b), and better opportunities for democratic participation beyond elections (Hypothesis 2c).

## Materials and methods

Study 2 used the same dataset and the same individual-level variables as in Study 1. Descriptive statistics for country-level variables are shown in Table 6.

**Country-level variables.** Economic development was measured by gross domestic product (GDP) per capita in 2012 expressed in thousands of US dollars (available in the ESS country dataset; we used a logged variable in regression analyses). Gini coefficient in 2012 represented income inequality (taken from [93]). Social welfare spending was a variable from the ESS country dataset, which measured social expenditure in 2010 (the most recent period available) in percent of GDP. Access to free elections was operationalized by electoral democracy index (covering freedom of association, clean elections, freedom of expression, elected officials, and suffrage), whereas participation opportunities were measured by participatory component index (covering civil society participation, elected local or regional government power, and direct popular vote), both referring to 2012 and provided by Varieties of Democracy Project (V-Dem) [94].

**Analytical approach.** We built on the same multilevel models as described in Study 1 [88]. As we obviously had low power at the between-country level (maximally 29 units), we proceeded in the following steps. First, we obtained estimates of the main effects of nonpolitical volunteering and political participation for each country from the random-intercepts-and-slopes models without country-level predictors. We plotted the distributions of these effects across countries and created scatterplots for each country-level predictor (X) and the effects of nonpolitical volunteering or political participation on each outcome considered (Y). Our purpose was to identify potential outliers among countries and to eyeball whether there were trends in expected directions. Second, we introduced country-level predictors of the random intercept (i.e., the main effect of a country-level variable on a well-being outcome) and slope (i.e., a cross-level interaction between the country-level variable and nonpolitical volunteering or political participation at the individual level) one by one to keep the number of parameters at the between-country level at a minimum. In case of significant cross-level interactions with

**Table 6. Descriptive statistics and correlations for the country-level predictors (Study 2).**

| Variable | N | M (SD) [a] | 1 | 2 | 3 | 4 | 5 |
|---|---|---|---|---|---|---|---|
| 1. GDP per capita 2012 | 29 | 32,893.35 (23,574.80) | – | | | | |
| 2. Gini coefficient 2012 | 28 | 31.16 (4.56) | -.26 | – | | | |
| 3. Social expenditure 2010 | 23 | 22.64 (4.60) | .10 | -.08 | – | | |
| 4. Electoral democracy index 2012 | 29 | 0.81 (0.16) | .55** | -.27 | .42* | – | |
| 5. Participatory component index 2012 | 29 | 0.66 (0.09) | .46* | -.28 | -.04 | .74** | – |

[a] Summary statistics across all countries.

* $p < .05$

** $p < .01$.

two or more country-level variables, we intended to extend the models by controlling these variables for one another. To combat the problem of multiple significance testing, we set the significance level for cross-level interactions at $p < .01$, but we considered trends under this threshold value as well—in case such trends were consistent across the outcomes—because our analyses with 29 countries were underpowered.

## Results

**Descriptive findings.** Fig 3A and 3B show the distributions of estimated main effects of nonpolitical volunteering and political participation on learning new things, as they are representative of the patterns described below (see S1 Fig for all outcomes). Inspection of these distributions suggested that the associations of nonpolitical volunteering with well-being were most favorable in some East European countries, including mainly Lithuania and Hungary (across outcomes), but also Bulgaria and Slovakia (for eudaimonic outcomes). The most unfavorable (or null) associations of nonpolitical volunteering were observed in some West or North European countries (e.g., Germany, the Netherlands, Norway, or Switzerland), with more variation across outcomes. There were some marked exceptions to this pattern: For instance, in Russia, nonpolitical volunteering had conspicuously unfavorable associations with sense of direction, generalized trust, and perceived social support, but this was not the case for other outcomes. Political participation had the most favorable associations with eudaimonic outcomes in some East European countries (especially Bulgaria, Estonia, Hungary, and Ukraine). At the same time, Russia (for all outcomes except for sense of competence and learning new things), but also Ukraine (for generalized trust and loneliness), featured the most unfavorable effects of political participation. In some countries (e.g., Cyprus), there were favorable associations of political participation with social but unfavorable associations with eudaimonic well-being outcomes. The East–West axis appeared to be much less prominent in the distribution of the associations of political participation with well-being than in those of nonpolitical volunteering.

We inspected the scatterplots of the main effects of nonpolitical volunteering and political participation on well-being outcomes against the country-level predictors (see S2 Fig). Virtually no systematic trends were recognizable except for GDP, which appeared to be inversely associated with the main effects of nonpolitical volunteering on eudaimonic well-being (see Fig 4 for flow experiences as an outcome, which is representative of this pattern). In line with Hypothesis 1a, the higher the country GDP, the less pronounced the positive associations appeared to be. (This applied even to outcomes that were not considered in our hypotheses for nonpolitical volunteering: sense of competence and learning new things; see S2 Fig.) Another apparent trend was observed for the associations of both nonpolitical volunteering and

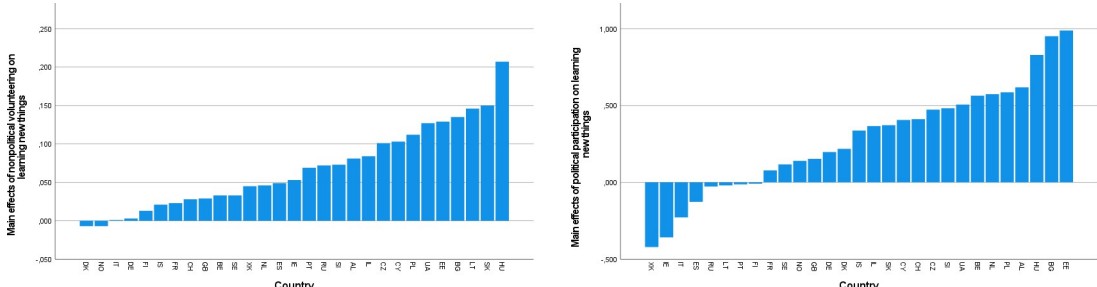

**Fig 3.** a. Country distribution of the main effects of nonpolitical volunteering on learning new things. b. Country distributions of the main effects of political participation on learning new things.

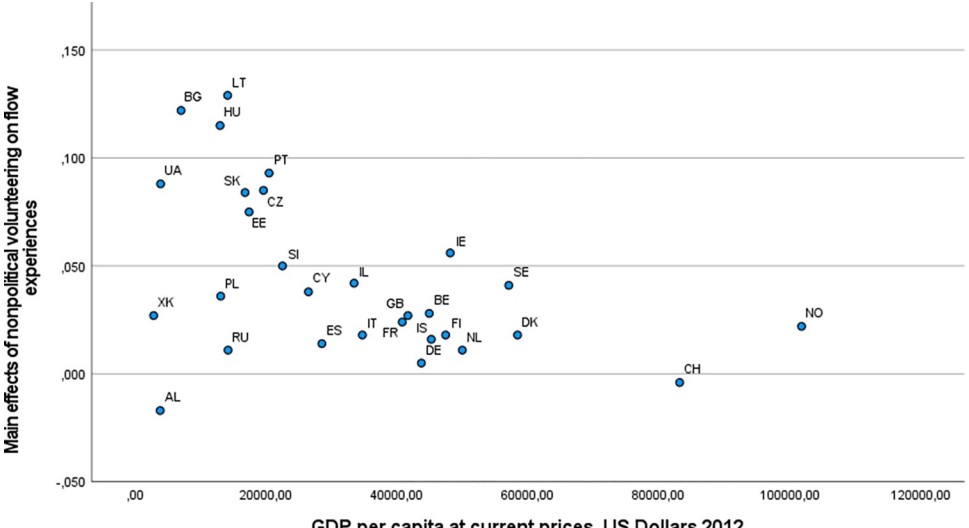

**Fig 4.** A scatterplot of the main effects of nonpolitical volunteering on flow experiences against GDP per capita.

political participation with generalized trust, which seemed to become more positive as the Gini coefficient increased; Russia was a major outlier in this pattern. This trend was in line with Hypothesis 1b but against Hypothesis 2a. The scatterplots also indicated that there was little variation on one country-level variable, electoral democracy index, with the vast majority of countries having an index above .80 (with 1.00 as the possible maximum).

**Findings from multilevel analyses.** Regression results for eudaimonic and social well-being are summarized in Tables 7 and 8, respectively. To take flow experiences as an example (see Table 7), the row "Intercept" shows, in the first column, the intercept (equaling to zero for a latent variable), and in the second column, the association of nonpolitical volunteering with flow experiences at average GDP per capita. This association is positive and significant. The next row contains the effects of GDP on flow experiences (random intercept) and on the random slope of nonpolitical volunteering (a two-way interaction). Whereas GDP has no main effect on flow experiences, it reduces the otherwise positive effect of nonpolitical volunteering (however, only at $p < .05$) and explains 10.9% of variance in this effect at the between-country level. The same pattern emerged for sense of direction (see Table 7; 8.9% of variance in random slope explained by GDP) but not for social support, although GDP had a significantly positive main effect on the latter (see Table 8). Thus, Hypothesis 1a was supported for two eudaimonic outcomes. As regards the Gini coefficient and social expenditure, neither had they main effects on the three outcomes tested nor did they modify the associations of nonpolitical volunteering with these outcomes. Thus, Hypotheses 1b and 1c were not supported.

For political participation, following results emerged. The Gini coefficient did not modify its associations with flow experiences, sense of competence, and generalized trust. However, it had marginally significant (at $p < .05$) effects on the random slope of political participation for perceived social support and loneliness, and these effects went in substantively opposite directions. Specifically, at average values of the Gini coefficient, political participation was negatively associated with perceived social support and positively associated with loneliness, which was consistent. However, with rising Gini values, the associations of political participation with both social support and loneliness became more negative, which was contradictory: The unfavorable effects on social support were thus exacerbated (in line with Hypothesis 2a), whereas the unfavorable effects on loneliness were weakened (against Hypothesis 2a). Gini

**Table 7. Country-level effects of contextual variables on eudaimonic well-being and cross-level interactions with voluntary participation.**

| | Flow experiences | $S_{\text{flow on nonpolitical volunteering}}$ | $S_{\text{flow on political participation}}$ | Sense of direction | $S_{\text{sense of direction on nonpolitical volunteering}}$ | Sense of competence | $S_{\text{sense of competence on political participation}}$ |
|---|---|---|---|---|---|---|---|
| Intercept | 0.00[a] | 0.05** (0.01) | | 6.91** (0.10) | 0.06** (0.02) | | |
| GDP per capita in 1000 USD, logged | -0.02 (0.07) | -0.02* (0.01) | | 0.08 (0.11) | -0.04* (0.02) | | |
| Intercept | 0.00[a] | 0.04** (0.01) | | 6.90** (0.10) | 0.05** (0.02) | | |
| Gini coefficient | 0.00 (0.01) | 0.00 (0.00) | | -0.02 (0.02) | 0.00 (0.00) | | |
| Intercept | 0.00[a] | | 0.09 (0.06) | | | 0.00[a] | 0.09 (0.08) |
| Gini coefficient | 0.00 (0.01) | | -0.03 (0.01) | | | -0.01 (0.02) | -0.02 (0.02) |
| Intercept | 0.00[a] | 0.05** (0.01) | | 6.91** (0.10) | 0.05** (0.02) | | |
| Social expenditure | 0.01 (0.01) | 0.00 (0.00) | | 0.01 (0.02) | 0.00 (0.00) | | |
| Intercept | 0.00[a] | | 0.08 (0.06) | | | 0.00[a] | 0.11 (0.07) |
| Electoral democracy index | -0.10 (0.38) | | 0.39 (0.42) | | | 0.96* (0.39) | -1.07* (0.46) |
| Intercept | 0.00[a] | | 0.09 (0.06) | | | 0.00[a] | 0.10 (0.08) |
| Participatory component index | -0.10 (0.65) | | 0.76 (0.72) | | | 1.50* (0.68) | -1.41 (0.85) |

*Note.* $N$ = 44,139–54,673. Reduced sample size in some models was due to the exclusion of countries with missing data on specific country-level indicators. Each pair of rows (intercept + predictor) and each pair of columns (outcome + slope) represent a separate regression model with one cross-level two-way interaction tested. Cells show unstandardized linear regression coefficients with standard errors in parentheses. Effects were adjusted for the full set of control variables at the individual level (age was included as a continuous covariate). Flow experiences and sense of competence were modeled as latent variables. S = random slope.

[a] The intercept of a latent variable is fixed at zero.

* $p < .05$

** $p < .01$.

coefficient accounted for 15.5% and 23.2% of between-country variance in the effects of political participation on social support and loneliness, respectively.

Electoral democracy index had no main effects on flow experiences or perceived social support, but it did on sense of competence and generalized trust (positive) and loneliness (negative). The only cross-level interaction effect (marginally significant at $p < .05$) emerged for sense of competence, whereby the association of political participation with this outcome was more negative in more democratic countries (13.6% of between-country variance in this effect was accounted for by electoral democracy index). Thus, Hypothesis 2b (more positive effects of political participation in countries with a better access to free elections) was rejected. Finally, participatory component index had main effects only on sense of competence (positive) and loneliness (negative) but not on the other outcomes tested. Contrary to Hypothesis 2c, it did not modify the effects of political participation.

**Post-hoc analyses.** As our initial inspection of scatterplots suggested that GDP might play a role in the associations of nonpolitical volunteering with all eudaimonic outcomes, we estimated the same models as above, with GDP as a country-level predictor, for sense of competence and learning new things, which were originally not included in our hypotheses. GDP per capita had positive main effects on both outcomes, $B$ ($SE$) = 0.17 (0.07), $p < .05$, and $B$ ($SE$) = 0.25 (0.07), $p < .01$, respectively. Furthermore, a significant interaction emerged between GDP per capita and nonpolitical volunteering for learning new things as outcome, $B$ ($SE$) = -0.04 (0.01), $p < .01$, but not for sense of competence, $B$ ($SE$) = -0.03 (0.02), *ns*. The significant interaction was consistent with previous findings: The higher the country-level GDP per capita, the less positive the associations of nonpolitical volunteering with learning new things.

**Table 8. Country-level effects of contextual variables on social well-being and cross-level interactions with voluntary participation.**

| | Generalized trust | $S_{generalized\ trust\ on\ political\ participation}$ | Perceived social support | $S_{social\ support\ on\ nonpolitical\ volunteering}$ | $S_{social\ support\ on\ political\ participation}$ | Loneliness | $S_{loneliness\ on\ political\ participation}$ |
|---|---|---|---|---|---|---|---|
| Intercept | | | 0.00[a] | 0.01 (0.01) | | | |
| GDP per capita in 1000 USD, logged | | | 0.18* (0.07) | -0.02 (0.01) | | | |
| Intercept | | | 0.00[a] | 0.01 (0.01) | | | |
| Gini coefficient | | | 0.00 (0.02) | 0.00 (0.00) | | | |
| Intercept | 0.00[a] | -0.04 (0.06) | 0.00[a] | | -0.16** (0.05) | 1.70** (0.04) | 0.10** (0.02) |
| Gini coefficient | -0.04 (0.03) | 0.03 (0.02) | 0.00 (0.02) | | -0.02* (0.01) | 0.01 (0.01) | -0.01* (0.01) |
| Intercept | | | 0.00[a] | 0.01 (0.01) | | | |
| Social expenditure | | | 0.01 (0.02) | 0.00 (0.00) | | | |
| Intercept | 0.00[a] | -0.04 (0.07) | 0.00[a] | | -0.17** (0.05) | 1.70** (0.03) | 0.10** (0.02) |
| Electoral democracy index | 1.81* (0.71) | 0.81 (0.43) | 0.82 (0.43) | | 0.61 (0.33) | -0.67** (0.17) | -0.06 (0.15) |
| Intercept | 0.00[a] | -0.03 (0.07) | 0.00[a] | | -0.16** (0.05) | 1.70** (0.03) | 0.10** (0.02) |
| Participatory component index | 2.12 (1.32) | 1.36 (0.74) | 1.14 (0.76) | | 0.61 (0.60) | -0.94** (0.32) | 0.08 (0.26) |

*Note.* $N$ = 44,139–54,673. Reduced sample size in some models was due to the exclusion of countries with missing data on specific country-level indicators. Each pair of rows (intercept + predictor) and each pair of columns (outcome + slope) represent a separate regression model with one cross-level two-way interaction tested. Cells represent unstandardized linear regression coefficients with standard errors in parentheses. Effects were adjusted for the full set of control variables at the individual level (age was included as a continuous covariate). Generalized trust and perceived social support were modeled as latent variables. S = random slope.

[a] The intercept of a latent variable is fixed at zero.

* $p < .05$

** $p < .01$.

Furthermore, as the scatterplots indicated that Russia might be an outlier in the relationship between the Gini coefficient, both types of voluntary participation, and generalized trust, we estimated the models for these variables without Russia. Marginally significant interactions emerged between the Gini coefficient and nonpolitical volunteering, $B$ ($SE$) = 0.004 (0.002), $p$ < .05, and between the Gini coefficient and political participation, $B$ ($SE$) = 0.039 (0.014), $p$ < .05. These effects confirmed the trends recognizable in the scatterplots: The associations of both types of voluntary participation with generalized trust became more positive with rising Gini values.

Finally, we compared the results from models with and without control variables at the individual level. The cross-level interactions with nonpolitical volunteering were significant in unadjusted models as well, whereas the interactions with political participation were not. This finding could probably be attributed to the (less favorable) individual-level associations of political participation with outcomes being found only in adjusted models in the first place (see Study 1). Furthermore, more (marginally) significant cross-level interactions emerged in unadjusted models. The moderating effect of GDP per capita on the association between non-political volunteering and perceived social support (in the same direction as for eudaimonic outcomes) was marginally significant, $B$ ($SE$) = -0.02 (0.01), $p$ < .05, $R^2$ = .096. Electoral democracy index and participatory component index moderated the association of political participation with generalized trust in expected direction (i.e., a more positive association in more democratic countries): $B$ ($SE$) = 1.14 (0.45), $p$ < .05, $R^2$ = .158, and $B$ ($SE$) = 1.87 (0.77), $p$ < .05, $R^2$ = .126, respectively. Electoral democracy index also moderated the association of

political participation with perceived social support in the same direction, $B$ ($SE$) = 0.81 (0.31), $p < .01$, $R^2$ = .232.

## Discussion

Study 2 showed that GDP per capita could account for some of the cross-national variation in the associations of nonpolitical volunteering with eudaimonic outcomes, those associations being more positive in the European countries with lower economic development. Thus, our argument that nonpolitical volunteering, which typically aims at direct helping on a small scale, may be more fulfilling where the need is greater received some empirical support. Economic inequality (the Gini coefficient) played a similar role in the relationship between nonpolitical volunteering and generalized trust (an outcome that was not foreseen by this hypothesis, though). We checked whether the moderating effects of GDP could in fact be attributed to lower volunteering rates in low-GDP countries [22, 27]. This was indeed the case; however, volunteering rates might mediate the effects of economic development, therefore it would be premature to discard the latter as an explanation for the variation in the effects of nonpolitical volunteering.

Our idea that political participation would be more difficult and conflict-laden and hence less rewarding in less democratic and more unequal societies was not substantiated by the data. In adjusted models, (marginally) significant cross-level interactions between country-level variables and political participation were either in the wrong direction or inconsistent. In unadjusted models only, several effects in the expected direction emerged, which indicated that the associations between political participation and social well-being (generalized trust in particular) were more positive in more democratic countries. As these effects disappeared after inclusion of control variables, this finding suggested that self-selection of more trusting individuals with better social relationships into political participation is stronger in more democratic countries. In well-functioning democracies, most political participation of citizens is part of the political system and aims not to overthrow the existing social order but to improve it. It stands to reason that citizens need trust in the system to participate in the democratic process. In contrast, much political activism in less democratic countries may be driven by distrust toward and dissatisfaction with the system [95].

To conclude, the only country-level variable that explained some of the variation in the effects of voluntary participation was GDP per capita, and this only for nonpolitical volunteering and eudaimonic well-being outcomes. In Study 3, we attempted to identify country-level variables that might be responsible for the cross-national variation in age differences in the effects.

## Study 3

In this study, we set out to explain why in some European countries, older adults apparently benefited from nonpolitical volunteering or political participation more than their younger counterparts did, whereas in other European countries, no age differences or differences in the opposite direction were observed (see a preregistration at https://osf.io/7ks45). Again, to reduce the number of cross-level interactions to be tested, we formulated hypotheses that were specific to certain outcomes or to a certain participation type. As Study 1 found that the associations between both types of voluntary participation and learning new things were more positive in older adults, we considered this outcome, along with sense of competence, as well-being indicators closely linked to individual productivity (in old age). We tested two alternative hypotheses here: In terms of sense of competence and learning new things, older adults might benefit from both nonpolitical volunteering and political participation more in countries with a *higher* effective retirement age, because of a stronger social norm to remain productive in old

age, with which engaged individuals comply (Hypothesis 1a) [96]. Alternatively, older adults might benefit more in countries with a *lower* effective retirement age, because engaged older individuals might feel superior to their age peers (Hypothesis 1b) [97].

Furthermore, we expected life expectancy at age 65 to differentially moderate the associations of voluntary participation with eudaimonic and social outcomes. In countries with higher life expectancy of the elderly, older adults have good access to healthcare and other resources and may feel a need to "fill time" in retirement (and to make up for the many previous years of toil [98]). In such a context, both nonpolitical volunteering and political participation may fill the activity gap well and may therefore feel especially fulfilling (i.e., where life expectancy at age 65 is higher, the associations with flow experiences and sense of direction are more positive in older adults; Hypothesis 2a). At the same time, in countries with lower life expectancy of the elderly, this age group experiences more losses in their social networks and has a lower access to resources, which puts them at risk of social isolation [99]. Voluntary participation may serve as a good compensation for such social losses, this is why we expected the associations of both nonpolitical volunteering and political participation with the quality of personal relationships (i.e., perceived social support and low loneliness) to be more positive in older adults in countries with lower life expectancy at age 65 (Hypothesis 2b).

The following hypotheses differentiated between the effects of nonpolitical volunteering and political participation. First, we considered the political power of youth, operationalized via the (comparatively low) mean age of members of parliament and cabinet. In countries where political elites are older, it may be difficult for young people to find their way into conventional political participation (as politics becomes increasingly professionalized [100]), and political activism may involve a lot of intergenerational conflict. We therefore hypothesized that in countries with a lower youth political power, social well-being outcomes of political participation would be more negative in younger people (Hypothesis 3a). In contrast, the learning effects of political participation in younger people may be stronger where few young activists face legions of mature politicians. In such country contexts, politically engaged youth can learn from intergenerational interactions and capitalize on their access to the networks of "old white men" (i.e., political elites [101]). We therefore expected that in countries with a lower youth political power, the associations of political participation with sense of competence and learning new things would be more positive in younger people (Hypothesis 3b).

Second, we considered the general economic situation of younger people in a given country, operationalized via youth unemployment rate. Drawing on our findings from Study 2, which suggested nonpolitical volunteering to be more rewarding where the need is greater, we hypothesized that nonpolitical volunteering would have more positive effects in younger people across all outcomes in countries with a higher youth unemployment rate (Hypothesis 4a). In such contexts, volunteering may be more needed by help recipients from one's own age group, compensate for poor employment opportunities, and provide access to social networks that may improve one's personal employment situation [cf. 102]. As regards political participation, we stuck to our argument that it might be less rewarding where it is more difficult. Redressing poor economic conditions is likely to be both a common and a very difficult target of youth political activism in countries affected by high youth unemployment [103]. Thus, across outcomes, we expected the effects of political participation to be less positive in younger people in countries where youth unemployment rate is higher (Hypothesis 4b).

## Materials and methods

Study 3 used the same dataset and the same individual-level variables as in Studies 1 and 2. Descriptive statistics for the country-level predictors are shown in Table 9.

**Table 9. Descriptive statistics and correlations for the country-level predictors (Study 3).**

| Variable | N | M (SD) [a] | 1 | 2 | 3 | 4 |
|---|---|---|---|---|---|---|
| 1. Effective retirement age | 28 | 63.24 (2.34) | – | | | |
| 2. Life expectancy at age 65 | 28 | 18.85 (1.73) | .39* | – | | |
| 3. Mean age MPs and cabinet | 29 | 48.08 (1.67) | -.22 | .01 | – | |
| 4. Youth unemployment rate | 29 | 20.28 (9.54) | -.29 | -.14 | -.01 | – |

[a] Summary statistics across all countries.

* $p < .05$

** $p < .01$.

**Country-level variables.** We used average effective retirement age for men (in years) between 2007 and 2012 [104], because more stable work biographies of men make their (higher) retirement age more indicative of the employment norm in old age. Average life expectancy at age 65 (in years) referred to 2012 [105]. To operationalize youth political power, we averaged mean age of members of the national parliament and mean age of cabinet members for each country, expressed in years [106]. Youth unemployment rate (in percent) referred to ILO unemployment estimates in 15–24-year-olds, averaged for the period between 2007 and 2012 [107]. For some countries, we found missing information on the above indicators in sources other than cited above (see the preregistration at https://osf.io/7ks45 for full details). We intended to use country-level variables from Study 2 (e.g., GDP per capita) as control variables if significant cross-level interactions with the focal country-level variables emerged.

**Analytical approach.** As in Study 2, we first inspected the distributions of estimated effects (here, interactions between nonpolitical volunteering/political participation and age groups) across countries and their scatterplots with the country-level variables. Then we used the same two-level setup as in Study 2, with the following adjustments: We estimated random slopes not only for the main effects of voluntary participation but also for the effects of age groups and for the interactions between voluntary participation and age. We regressed these random slopes, along with the outcome (random intercept), on the country-level variables. In each model, only one country-level variable was included to keep the number of parameters at the between-country level at a minimum. We applied cluster-mean centering to the indicators of voluntary participation before computing their interactions with age. To facilitate interpretation, neither age group dummies nor the interaction effects were further centered (for all outcomes, the variance of these variables at the between-country level was below 1% of their total variance). For more details about the analysis plan, see Studies 1 and 2 and the preregistration at https://osf.io/7ks45.

## Results

### Descriptive findings

Fig 5A and 5B show the distributions of the interactions between nonpolitical volunteering and age groups (younger vs. older; middle-aged vs. older) for flow experiences (see S3 Fig for all distributions). In some countries, the associations of nonpolitical volunteering with well-being were more positive in younger than in older adults (e.g., in the UK and Spain for several eudaimonic and social outcomes; in Iceland for eudaimonic outcomes; and in Albania for social outcomes). In contrast, in several East and South European countries (e.g., in Russia for almost all outcomes, in Kosovo and Israel for eudaimonic outcomes, and in Czech Republic

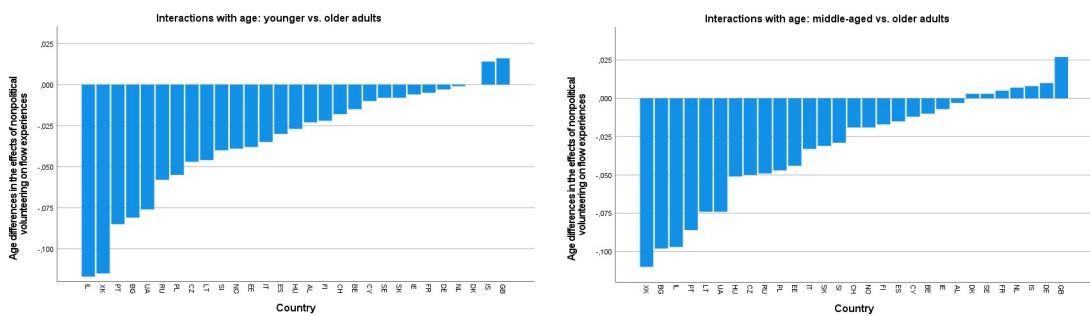

**Fig 5.** a. Country distribution of the interactions between nonpolitical volunteering and age (younger vs. older adults) in relation to flow experiences. b. Country distribution of the interactions between nonpolitical volunteering and age (middle-aged vs. older adults) in relation to flow experiences.

for social outcomes), these associations were more unfavorable in younger than in older adults. Regarding the contrast between middle-aged and older adults, the associations of nonpolitical volunteering with eudaimonic well-being (excepting sense of direction) appeared to be in favor of middle-aged adults in some West and North European countries (e.g., Germany, Iceland, and Sweden) and in favor of older adults in some East and South European countries (e.g., Kosovo, Russia, and Israel). For sense of direction and social well-being, there was no recognizable pattern of country differences in this contrast.

Virtually across outcomes, the associations of political participation with well-being were more favorable in younger than in older adults in the UK (see Fig 6A and 6B for the distributions of interaction effects for perceived social support and S3 Fig for other outcomes). It was the other way round in several East and South European countries, especially Israel (for eudaimonic outcomes), Bulgaria, Estonia, and Portugal (across outcomes), and Russia (particularly for perceived social support). Political participation had more favorable effects in middle-aged than in older adults in Spain (most notably for sense of direction and perceived social support) and less favorable effects in middle-aged than in older adults in Israel (for eudaimonic outcomes) and in Albania and Portugal (almost across outcomes). No clear differentiation along the East–West or North–South axis was recognizable here.

Inspection of scatterplots suggested that the difference between younger/middle-aged and older adults in the associations of nonpolitical volunteering with flow experiences was more positive (i.e., in favor of younger individuals) in countries with a longer life expectancy at age 65 (see Fig 7A and 7B). Israel and Portugal represented outliers in this pattern. This trend ran

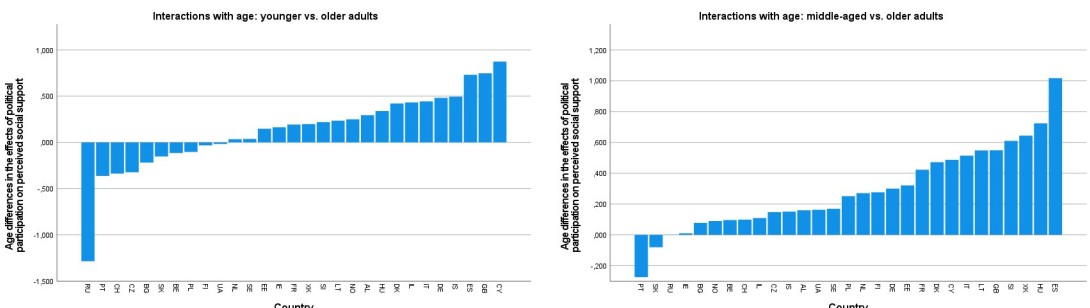

**Fig 6.** a. Country distribution of the interactions between political participation and age (younger vs. older adults) in relation to perceived social support. b. Country distribution of the interactions between political participation and age (middle-aged vs. older adults) in relation to perceived social support.

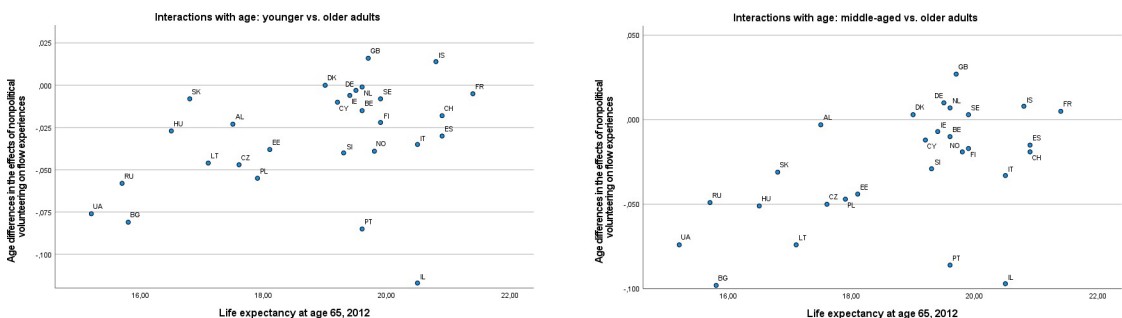

**Fig 7.** a. A scatterplot of the interactions between nonpolitical volunteering and age (younger vs. older adults) against life expectancy at age 65 for flow experiences as an outcome variable. b. A scatterplot of the interactions between nonpolitical volunteering and age (middle-aged vs. older adults) against life expectancy at age 65 for flow experiences as an outcome variable.

against Hypothesis 2a. Furthermore, as regards the associations of political participation with sense of competence and learning experiences, the scatterplots indicated that in three countries with especially high mean age of members of parliament and cabinet (i.e., the UK, France, and Italy), these associations were more positive in younger than in older adults (see Fig 8 for learning new things and S4 Fig for all scatterplots). In contrast, in countries with especially low mean age of political elites (e.g., Bulgaria and Ukraine), the associations of political participation with these outcomes were more negative in younger than in older adults. Although this pattern was in line with Hypothesis 3b, no linear trend was present. There were no other recognizable trends or patterns in the scatterplots (see S4 Fig); thus, our hypotheses seemed unlikely to be supported.

**Findings from multilevel analyses.**   A summary of multilevel regression results with three-way cross-level interactions between country-level variables, indicators of voluntary participation, and age groups is shown in S2 and S3 Tables. Only isolated main effects of country-level variables on the outcomes emerged: Life expectancy at age 65 had significantly positive

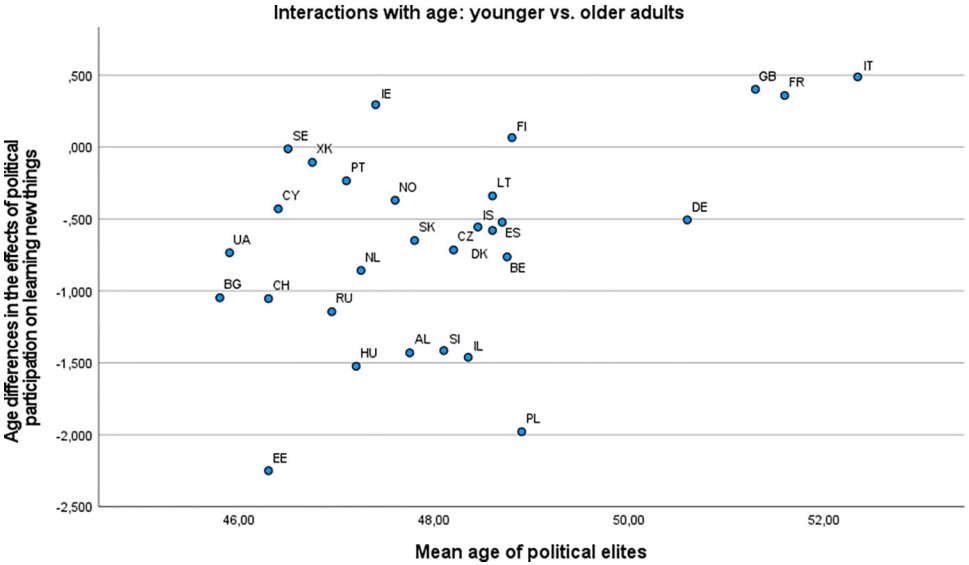

**Fig 8. A scatterplot of the interactions between political participation and age (younger vs. older adults) against mean age of members of the national parliament/cabinet for learning new things as an outcome variable.**

effects on flow experiences and perceived social support as well as a negative effect on loneliness, mean age of political elites had a positive effect on learning new things, and youth unemployment rate had a negative effect on generalized trust. A marginally significant (at $p < .05$) two-way cross-level interaction between life expectancy at age 65 and nonpolitical volunteering indicated that volunteering had less positive associations with flow experiences in countries with a higher life expectancy. No significant three-way cross-level interactions (not even at $p < .05$) emerged; thus, none of our hypotheses was supported.

Analyses without control variables at the individual level did not yield substantially different results. Post-hoc analyses with exclusion of certain countries for certain models (i.e., those where trends were present in the scatterplots, as described above) did not produce any more significant cross-level interactions either. Finally, as GDP per capita emerged as a major country-level explanatory variable in Study 2, we checked whether it might also explain the cross-national variation in the interactions with age. Again, we created scatterplots (see S5 Fig), which suggested that for flow experiences and sense of competence, but not for other eudaimonic or social well-being outcomes, the effects of both nonpolitical volunteering and political participation were more in favor of younger/middle-aged adults (in comparison to older adults) in countries with a higher GDP per capita. Multilevel analyses supported this trend only for sense of competence, where marginally significant (at $p < .05$) three-way cross-level interactions emerged (available upon request). Overall, these findings yielded little evidence for economic development playing a relevant role in these associations.

## Discussion

Study 3 revealed a large heterogeneity in age differences in the associations of voluntary participation with eudaimonic and social well-being (i.e., whether younger or older adults benefit from participation more) across European countries. However, we failed to identify country-level factors that would be responsible for this variation. This null finding was most likely not due to lacking statistical power to detect cross-level three-way interactions, because our inspection of scatterplots did not yield any systematic trends across outcomes either. Thus, cross-national differences in age differences may have to do with additional factors we did not consider, such as typical activities of younger, middle-aged, and older volunteers in a given country or the success and popularity of the country's largest voluntary organizations that target specific age groups.

## General discussion

The central finding of the present studies is that the associations of volunteering and political participation with eudaimonic and social well-being exhibit remarkable heterogeneity across European countries, which results in very small average effect sizes. Moreover, we showed that there is no general rule as regards age differences in these associations. There are countries where effects are more favorable in younger adults and others where effects are more favorable in older adults. Still other countries exhibit no clear patterns of age differences. Although we were unable to identify country-level factors that might be responsible for such variation, we know which variables are not: Effective retirement age in men, life expectancy at age 65, average age of members of the national parliament and cabinet, and youth unemployment rate were equally unsuitable to explain the cross-national variation in age differences. In contrast, the variation in main effects (especially in the associations of nonpolitical volunteering with eudaimonic well-being) was partly explained by economic development and further by volunteering rates. Specifically, volunteers appeared to experience more positive well-being outcomes in countries with lower economic development and lower volunteering rates.

Unexpectedly, the level of democracy played no systematic role in the link between political participation and well-being. Just as many European nations appear to be relatively homogenous and distinguishable in terms of central values they hold [108], they may have developed distinct participation cultures that account for the unique patterns of associations between voluntary participation and well-being that we found in the present studies.

Here are some country examples. In Russia, the estimated effects of political participation in particular were unfavorable almost across well-being outcomes, and this mainly pertained to younger engaged individuals. Of all European countries considered, Russia was the country with the lowest scores on democracy [94]. In 2012, first mass demonstrations against the ruling party and the president took place in large cities, which were partly violently suppressed. Even though the index of political participation used in the ESS might as well capture activism of pro-government youth movements such as "Nashi" ("ours"), in Russia, involvement in both oppositional and pro-government movements may entail exposure to interpersonal and intergroup hostility and violence. This may explain the negative effects of political participation in youth, who are often at the forefront of social movements [109]. This adverse feedback loop discouraging young Russians from political participation provides a hint to the developments of the subsequent years, which have seen the reinforcement of an authoritarian system, culminating in extraordinary political repressions during the war in Ukraine.

Israel is a country where the associations of voluntary participation with eudaimonic well-being were especially favorable in older adults, whereas in younger adults, political participation was associated with poor eudaimonic outcomes. Israel is a democratic country, which, however, is involved in a longstanding military conflict (with religious underpinnings) with Palestine. Probably many politically active young Israelis grapple with this conflict, which may engender feelings of immense frustration and hopelessness. As regards older Israelis, much of their voluntary participation may be driven by religious values and take place in religious communities, which may contribute to their positive experiences [61, 110, 111].

In Bulgaria, we found generally positive associations of both types of voluntary participation with well-being outcomes, especially eudaimonic well-being, and more favorable effects in older adults, especially as regards political participation. Similarly to Russia, Bulgaria is a post-socialist country, but unlike Russia, it is a EU member since 2007. At the end of 2000s, its civil society was characterized by extremely low levels of volunteering and political engagement, low trust in institutions and voluntary organizations, low public impact of NGOs, and even cases of corruption among the NGOs struggling to obtain EU funding [112]. At the same time, there was undoubtedly much need for voluntary action (e.g., high poverty and unemployment levels) and no substantial state interference with citizen's initiatives [112]. The few engaged individuals might be strongly values driven and therefore enjoy more benefits of participation than observed elsewhere. Notably, retirees (along with students and environmental activists) were perceived by general public as one of the groups authentically representing the Bulgarian civil society [112]. Thus, our finding of stronger positive effects in this age group might be attributed to a particular cultural and historical background of older Bulgarians' voluntary participation.

The UK stood out as a country where the associations of voluntary participation, especially political participation, with both eudaimonic and social well-being were more favorable in younger (and sometimes in middle-aged) than in older adults, whereas no apparent benefits were observed in the latter (cf. [10], who found small positive effects of political participation in the UK using large-scale panel data). The UK is an established democracy with a stable political system, and at least since the late 1990s, it has been placing a high value on the political socialization of youth (e.g., active citizenship education programs [113]). Because of the majoritarian electoral system, political careers in the UK typically start through engagement in

local communities [114]. At least before the Brexit turmoil, the UK society might have been successful in integrating new generations of young people into the existing political structures while giving them an opportunity to exercise initiative and promote change.

As a last example, in Germany, no systematic pattern emerged, with many effects being close to zero or showing opposing directions of age differences for different outcomes. Lühr et al. [9] reported similarly inconsistent and largely null effects of both types of voluntary participation for this country. Germany may be a typical "average case", where different institutional and historical factors work in opposing directions to produce highly heterogeneous effects of voluntary participation. Germany is a democracy (but with the tragic Nazi and GDR past), in which the state fosters (also financially) various nonpolitical and political voluntary organizations while retaining generous levels of social welfare spending (the corporatist model in social origins theory [115]). In contrast to Anglo-Saxon countries, Germany has no strong tradition of service learning or active citizenship education, although targeted volunteering programs for both youth and retirees are available. In this country context, voluntary participation may be seen as one lifestyle choice among others, a nice thing to have but not a must.

### Strengths and limitations

Our studies had a number of strengths. We used one of the largest cross-national comparative datasets available for studying the relationship between voluntary participation and well-being. A highly differentiated assessment of well-being outcomes, which were presumably very proximal to voluntary participation, added validity to our findings. Moreover, we preregistered our choice of measures and statistical analyses for each study. Still, there were inevitable limitations. The data stemmed from 2012, ESS Round 6, because the comprehensive well-being module has not been repeated since then. Afterwards, a lot has changed in the political and economic situation of the European countries, the most recent (and adverse) developments including the Brexit, the COVID-19 pandemic, and the war in Ukraine. It is reasonable to suppose, though, that our central finding still holds: The link between voluntary participation and eudaimonic and social well-being as well as age differences in this link vary strongly across countries.

On a more technical side, a cross-sectional design precluded us from making causal or directional inferences, although we attempted to combat this limitation by using a very extensive set of control variables. Furthermore, we used single-item measures to assess the frequency of nonpolitical volunteering and some of the well-being outcomes. Although we distinguished between nonpolitical volunteering and political participation, no information was available on specific activities or tasks undertaken (especially during volunteering). Our definition of age groups might be criticized as arbitrary or too broad in terms of age range within each group. However, most age-related hypotheses in the literature on voluntary participation focus on contrasting younger and older adults; indeed, in the present studies, where age differences emerged, the effects in middle-aged adults appeared to be in between. Besides, a more fine-grained differentiation by age combined with very low voluntary participation rates in some of the ESS countries could have led to extremely small cell sizes. Regarding the sample size, it was definitely small at the country level (29 countries), and this limitation was exacerbated by unavailability of data on some country-level indicators for some countries. It cannot be ruled out that, if full data were available, more significant effects of country-level predictors would have emerged. In most of the cases, however, inspecting the scatterplots showed that there were not even trends in expected directions. Finally, our choice of country-level indicators, although one of the most comprehensive to date in this field, was still limited in terms of availability of data, variation across countries (e.g., there was little variation in the electoral democracy index), and proximity to our research questions.

## Conclusions and future directions

In the present studies, we utilized cross-sectional data from Round 6 (2012) of the European Social Survey (ESS) to investigate cross-national differences in the associations of two indicators of voluntary participation, nonpolitical volunteering and political participation, with various dimensions of eudaimonic and social well-being. Furthermore, we addressed age differences in these associations and whether and why older or younger adults might benefit more in some European countries than in others. The main lessons learned are as follows. First, on average across European countries, nonpolitical volunteering and political participation had negligibly small associations with eudaimonic and social well-being, with few age differences detected. Second, on average and across well-being outcomes, the effects of nonpolitical volunteering were significantly more positive than those of political participation, providing some support for our argument that political participation may be more difficult and conflict-prone. Third, there was considerable variation across European countries in the direction and size of these effects (and in whether younger or older adults appeared to benefit more). Fourth, a small part of the variation in main effects was explained by GDP per capita, with nonpolitical volunteering having more positive associations with eudaimonic well-being in countries with lower GDP, a finding that we attribute to volunteers benefiting from their engagement more where the need is greater. Fifth, we failed to identify country-level variables that could account for the variation in the effects of political participation or in age differences in the effects of both indicators of voluntary participation.

To gain more insight into the sources of cross-national differences in the effects of voluntary participation on well-being, future researchers will need more specific information on the country context (e.g., major voluntary organizations active in a given country or major areas of voluntary participation; public attitudes toward volunteering and political participation) and on activities undertaken by engaged individuals. The idiosyncrasies of cultures, such as cultural values, might also play a role, although the distributions of the effects of voluntary participation across the ESS countries did not appear to map onto particular dimensions such as traditional versus secular values or power distance [116, 117]. Furthermore, it might be more promising to focus on the cross-national differences in the immediate subjective outcomes of voluntary participation (e.g., quality of experience) rather than in the overall measures of subjective, eudaimonic, or social well-being, as was done in the present and in many preceding studies. Finally, although difficult to implement, collecting data from many more countries and, ideally, using repeated measurements would provide decisive methodological advances to push this research field further.

## Supporting information

**S1 Table. Country-level variance estimates for the main effects of voluntary participation on eudaimonic and social well-being and for the interactions with age (Study 1).**
(PDF)

**S2 Table. Cross-level three-way interactions between contextual variables, voluntary participation, and age for eudaimonic well-being (Study 3).**
(PDF)

**S3 Table. Cross-level three-way interactions between contextual variables, voluntary participation, and age for social well-being (Study 3).**
(PDF)

**S1 Fig. Country distributions of estimated main effects of voluntary participation on all eudaimonic and social well-being outcomes (Study 2).**
(PDF)

**S2 Fig. Scatterplots of the main effects of voluntary participation all eudaimonic and social well-being outcomes against each country-level indicator (Study 2).**
(PDF)

**S3 Fig. Country distributions of the interactions between voluntary participation and age in relation to eudaimonic and social well-being outcomes (Study 3).**
(PDF)

**S4 Fig. Scatterplots of the interactions between voluntary participation and age against all country-level indicators foreseen by the hypotheses (Study 3).**
(PDF)

**S5 Fig. Scatterplots of the interactions between voluntary participation and age against GDP per capita (Study 3, post-hoc analyses).**
(PDF)

## Acknowledgments

We thank John Wilson and Jane Piliavin for their helpful comments on the grant proposal, Elena Semenova for her useful advice on the country-level variables relevant to political participation as well as for her comments on the country portraits, and Inga Ungruh for her assistance in formatting figures.

## Author Contributions

**Conceptualization:** Maria K. Pavlova.

**Formal analysis:** Matthias Lühr.

**Methodology:** Maria K. Pavlova, Matthias Lühr.

**Project administration:** Maria K. Pavlova.

**Supervision:** Maria K. Pavlova.

**Writing – original draft:** Maria K. Pavlova.

**Writing – review & editing:** Maria K. Pavlova, Matthias Lühr.

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
