## [Decision Letter · Decision Letter 0]

6 Dec 2022

PONE-D-22-20390Volunteering and Political Participation Are Differentially Associated With Eudaimonic and Social Well-Being Across Age Groups and European CountriesPLOS ONE

Dear Dr. Maria Pavlova,

Thank you for submitting your manuscript to PLOS ONE. After careful consideration, we feel that it has merit but does not fully meet PLOS ONE’s publication criteria as it currently stands. Therefore, we invite you to submit a revised version of the manuscript that addresses the points raised during the review process.

Dear Maria Pavlova  I write you in regards to a manuscript entitled "Volunteering and Political Participation Are Differentially Associated With Eudaimonic and Social Well-Being Across Age Groups and European Countries" which you submitted to Plos One. After reading your paper and receiving two reports about the paper I have decided to allow the adaptation of the paper to the suggestions of the both referees (Major Revision.

We look forward to receiving your revised manuscript.

Kind regards,

Maximo Rossi, PhD Economics

Academic Editor

PLOS ONE

Journal Requirements:

“This work was supported by a German Research Foundation (DFG) grant (grant number PA 2704/3-1) to the first author.”

5. Please include captions for your Supporting Information files at the end of your manuscript, and update any in-text citations to match accordingly. Please see our Supporting Information guidelines for more information: http://journals.plos.org/plosone/s/supporting-information

Additional Editor Comments (if provided):

Dear Editor, the paper constitutes a potential publication as long as it raises a series of observations that I support. Best. Maximo

Reviewers' comments:

Reviewer's Responses to Questions

**Comments to the Author**

1. Is the manuscript technically sound, and do the data support the conclusions?

Reviewer #1: Partly

Reviewer #2: Yes

2. Has the statistical analysis been performed appropriately and rigorously? 

Reviewer #1: No

Reviewer #2: I Don't Know

3. Have the authors made all data underlying the findings in their manuscript fully available?

Reviewer #1: No

Reviewer #2: Yes

4. Is the manuscript presented in an intelligible fashion and written in standard English?

Reviewer #1: Yes

Reviewer #2: Yes

5. Review Comments to the Author

Reviewer #1: The topic of the paper is quite interesting although not original. There is a huge literature both on the effects and on the determinants of volunteering.

It is not clear whether the authors are studying the effects of volunteering or the reasons why people and especially the elderly decide to volunteer. Therefore, it could be useful providing the reader with this kind of information within a better specified background of the literature.

In my opinion the research questions are too many making confusion about the aim of the paper: there are three papers in one. Consequently, the paper becomes very long and quite hard to read till the end. I suggest simplifying the goal.

Pag 3: the authors talk about formal volunteering and they juxtapose this activity with conventional political participation. I suggest mentioning that there exists also informal volunteering clarifying what it is and why conventional political participation cannot be considered as informal volunteering. However, I am not sure that conventional political participation is not a kind of formal volunteering, please could you better specify this point.

Pag 4: when the authors talk about “other well-being outcomes”, they do not consider that there is a very large literature on the effects of volunteering on health.

Page 5: the section “Eudaimonic Well-Being” seems quite confusing. There is a large literature that studies the reasons why people decide to volunteer. In the section the determinants of volunteering are managed as consequences of volunteering. So it could be useful to better specify for instance that young people volunteer to improve their skills or it could be the other way around, i.e., “that civic engagement was shown to improve academic, personal, and social competence”. In the “Social Well-Being” section, again, it is not clear whether the authors talk about the reasons why people volunteer or about their consequences.

Data: it is not clear why the authors use the European Social Survey, 2014 if it is available Round ten data. The data employed are quite old, the authors should explain why they use them.

Page 15: since the countries sample are numerically different, why do not you use weights for countries?

Weights seem to be necessary to provide each country with the same importance within the whole sample.

You said “Moreover, we could not use weighting to 7 correct for unequal selection probabilities as it is not available with Bayesian estimation 8 (Muthén & Muthén, 2017).” If this is the case you should better explain the reason, please.

Page 15: For missing values you need imputation. You should better explain the following: “Missing values on both dependent and independent variables at 9 the individual level were estimated directly in each model under MAR assumption”.

Page 16: “Multiple Regression Results”. Are you sure that you can talk about effects? Don’t you have just correlation?

Reviewer #2: overall i like the writing, clear and to the point

great topic; useful to break down by nonpolitical vol, and by country; i like

the corss country stuff; right--(effective

retirement age in men, life expectancy at 65, average age of members of the national

parliament and cabinet, and youth unemployment rate)--what about culture? say

https://en.wikipedia.org/wiki/Inglehart%E2%80%93Welzel_cultural_map_of_the_world

https://en.wikipedia.org/wiki/Hofstede%27s_cultural_dimensions_theory

" self-realization (eudaimonic well-being; Ryan & Deci, 2001)"

"quality of social relationships (social well-being; Keyes, 1998)."

--very good!

"Discrepant findings from different countries may result from random sampling

errors"--may but unlikely! any particular evidence of that?

and then you enumerate stuff, good, but again what about culture!

"Plagnol and Huppert (2010) found that formal volunteering was more

10

1 strongly associated with higher SWB, sense of accomplishment, and feelings of doing

2 something worthwhile in European countries with low (vs. high) volunteering rates."

--seems like an important study that you may follow up on and update

extra points for using linux! :)

i refrain from commenting on method as i dont use it

control variables look good

my only major criticism is that it's too long! see if you can move say 20-80perc of it into appendix/supplementary material

alternatively this could be 2 papers--one on age, one on cross-country differences

6. PLOS authors have the option to publish the peer review history of their article (what does this mean?). If published, this will include your full peer review and any attached files.

Reviewer #1: No

Reviewer #2: No

---

## [Author Response · Author response to Decision Letter 0]

22 Dec 2022

RESPONSE TO THE COMMENTS MADE BY THE EDITOR AND THE REVIEWERS

EDITOR

“Both referees observe the extension of the paper and the possibility of reducing the set of objectives.”

Our manuscript is a classical multi-study paper, whereby each next study builds on the previous one. That is, both Study 2 (cross-national differences in the main effects of volunteering and political participation) and Study 3 (cross-national differences in the age differences in these effects) build on Study 1, which shows that (a) contrary to a widespread opinion, there were few age differences in the effects of volunteering and political participation, and (b) there was a large cross-national variation in both main effects and age differences. Study 2 then investigates country-level factors that might explain the variation in the main effects, whereas Study 3 investigates different country-level factors that might explain the variation in age differences. Study 3 additionally uses some of the country-level variables from Study 2 as control variables or as additional predictors for supplementary analyses (GDP per capita). Splitting the studies in one way or another would, first, constitute a piecemeal publication, and second, effectively increase the overall volume of the manuscripts, because more information from the studies published separately (especially Study 1) would have to flow into the description of the other studies (especially Studies 2 and 3).

As far as we understand, Plos ONE is committed to publish preregistered research, especially with null or controversial findings, and imposes no limitations on the length of submitted manuscripts. One of publication criteria in Plos ONE is “Experiments, statistics, and other analyses are performed to a high technical standard and are described in sufficient detail”. We went through our manuscript and see no room for removing parts of study descriptions, such as Analytical Approach or Results, into supplementary information, because the descriptions in the manuscript are essential for understanding of our methodological approach and findings. (Besides, Plos ONE is an online-only publication, which makes a distinction between the main article and supplementary information even fuzzier.) There are fresh examples of multi-study papers in psychology published in Plos ONE, which have comparable length to our manuscript:

https://journals.plos.org/plosone/article?id=10.1371/journal.pone.0270225

https://journals.plos.org/plosone/article?id=10.1371/journal.pone.0268713

https://journals.plos.org/plosone/article?id=10.1371/journal.pone.0261251

At any rate, in the revised version, the word count was reduced by converting in-text references into numbers, as required by Plos style.

REVIEWER 1

We thank the reviewer for their helpful comments and suggestions.

“The topic of the paper is quite interesting although not original. There is a huge literature both on the effects and on the determinants of volunteering.”

We cite much of this literature in the literature review and clearly delineate our contribution to this body of research: we focused on the outcomes that have been comparatively rarely addressed in the literature (eudaimonic and social well-being) and on the variation in effects across types of participation (most literature considered volunteering without any further differentiation), age groups, and European countries. 

“It is not clear whether the authors are studying the effects of volunteering or the reasons why people and especially the elderly decide to volunteer. Therefore, it could be useful providing the reader with this kind of information within a better specified background of the literature.”

“Page 5: the section “Eudaimonic Well-Being” seems quite confusing. There is a large literature that studies the reasons why people decide to volunteer. In the section the determinants of volunteering are managed as consequences of volunteering. So it could be useful to better specify for instance that young people volunteer to improve their skills or it could be the other way around, i.e., “that civic engagement was shown to improve academic, personal, and social competence”. In the “Social Well-Being” section, again, it is not clear whether the authors talk about the reasons why people volunteer or about their consequences.”

In the revised manuscript (p. 6), we write:

“It should be noted that many of these outcomes can be simultaneously considered as incentives or motivations to volunteer, which have been investigated in a separate line of research literature (including cross-national differences therein; Hustinx et al., 2010). There is no contradiction, however, in that individuals volunteer in order to gain something, be it meaning of life or social connections, and that volunteering may indeed be instrumental in gaining that something. In the present paper, we focus on this latter aspect: documented outcomes of volunteering.”

In addition, we went through the literature review and ensured that it is clear everywhere that we write about the outcomes of volunteering. We did not review the literature on the motivations to volunteer in any detail because this would go beyond the scope of this paper.

“Pag 3: the authors talk about formal volunteering and they juxtapose this activity with conventional political participation. I suggest mentioning that there exists also informal volunteering clarifying what it is and why conventional political participation cannot be considered as informal volunteering. However, I am not sure that conventional political participation is not a kind of formal volunteering, please could you better specify this point.”

We juxtapose nonpolitical formal volunteering with political participation, which can of course include formal volunteering for political purposes. In the revised manuscript (p. 4), we write:

“Prior research has usually implied nonpolitical formal volunteering, which is an organized voluntary activity to help others (or to achieve another common good, such as cleaning a river bank) directly (Verba et al., 1995; J. Wilson, 2012). We juxtaposed this activity with conventional political participation: attempts to influence policy at different levels, which include not only political volunteering but also actions such as attending a demonstration (Verba et al., 1995). Although political participation is likewise directed at the common good, namely policy change or social change, it may be more difficult and conflict prone than is nonpolitical volunteering (Pavlova et al., 2021; Theiss-Morse & Hibbing, 2005). Both kinds of voluntary participation can be sometimes difficult to differentiate from informal volunteering, which is not attached to a formal organization but still involves unpaid voluntary work (e.g., providing help to nonkin; Wilson, 2012). However, the standard indices of volunteering and political participation that we used in the present studies referred explicitly to formal volunteering for nonpolitical or political purposes (i.e., “for organizations”) or listed actions that cannot be considered as unpaid work (e.g., “contacted a politician” or “signed a petition”) but qualify as political participation.”

“Pag 4: when the authors talk about “other well-being outcomes”, they do not consider that there is a very large literature on the effects of volunteering on health.”

In the revised manuscript (p. 5), we write:

“Subjective well-being (SWB, a hedonic aspect of well-being; Diener, 1994) has been the most studied well-being outcome of volunteering (Anderson et al., 2014), apart from its well-established effects on physical health (Anderson et al., 2014; Filges, 2020; Piliavin & Siegl, 2015).”

“In my opinion the research questions are too many making confusion about the aim of the paper: there are three papers in one. Consequently, the paper becomes very long and quite hard to read till the end. I suggest simplifying the goal.”

Our manuscript is a classical multi-study paper, whereby each next study builds on the previous one. That is, both Study 2 (cross-national differences in the main effects of volunteering and political participation) and Study 3 (cross-national differences in the age differences in these effects) build on Study 1, which shows that (a) contrary to a widespread opinion, there were few age differences in the effects of volunteering and political participation, and (b) there was a large cross-national variation in both main effects and age differences. Study 2 then investigates country-level factors that might explain the variation in the main effects, whereas Study 3 investigates different country-level factors that might explain the variation in age differences. Study 3 additionally uses some of the country-level variables from Study 2 as control variables or as additional predictors for supplementary analyses (GDP per capita). Splitting the studies in one way or another would, first, constitute a piecemeal publication, and second, effectively increase the overall volume of the manuscripts, because more information from the studies published separately (especially Study 1) would have to flow into the description of the other studies (especially Studies 2 and 3).

As far as we understand, Plos ONE is committed to publish preregistered research, especially with null or controversial findings, and imposes no limitations on the length of submitted manuscripts. One of publication criteria in Plos ONE is “Experiments, statistics, and other analyses are performed to a high technical standard and are described in sufficient detail”. We went through our manuscript and see no room for removing parts of study descriptions, such as Analytical Approach or Results, into supplementary information, because the descriptions in the manuscript are essential for understanding of our methodological approach and findings. (Besides, Plos ONE is an online-only publication, which makes a distinction between the main article and supplementary information even fuzzier.) There are fresh examples of multi-study papers in psychology published in Plos ONE, which have comparable length to our manuscript:

https://journals.plos.org/plosone/article?id=10.1371/journal.pone.0270225

https://journals.plos.org/plosone/article?id=10.1371/journal.pone.0268713

https://journals.plos.org/plosone/article?id=10.1371/journal.pone.0261251

At any rate, in the revised version, the word count was reduced by converting in-text references into numbers, as required by Plos style.

“Data: it is not clear why the authors use the European Social Survey, 2014 if it is available Round ten data. The data employed are quite old, the authors should explain why they use them.”

The extensive personal and social well-being module has been included into ESS in 2006/2007 and 2012/2013, and we used the later data. More recent modules include only very basic questions on hedonic well-being, which we do not use at all in our studies, because there is abound research on the effects of volunteering and political participation on such indicators (we cite some of it in the literature review). See here a description of ESS well-being measures:

http://www.esswellbeingmatters.org/measures/index.html

In our manuscript (pp. 5, 54), we explain:

“...We therefore focused on eudaimonic and social well-being and considered their specific dimensions, which may yield differential effects of nonpolitical volunteering and political participation. For this reason, we used ESS data from 2012, the latest round when a comprehensive personal and social well-being module was administered.”

“The data stemmed from 2012, ESS Round 6, because the comprehensive well-being module has not been repeated since then. Afterwards, a lot has changed in the political and economic situation of the European countries, the most recent (and adverse) developments including the Brexit, the COVID-19 pandemic, and the war in Ukraine. It is reasonable to suppose, though, that our central finding still holds: The link between voluntary participation and eudaimonic and social well-being as well as age differences in this link vary strongly across countries”

“Page 15: since the countries sample are numerically different, why do not you use weights for countries?

Weights seem to be necessary to provide each country with the same importance within the whole sample.

You said “Moreover, we could not use weighting to 7 correct for unequal selection probabilities as it is not available with Bayesian estimation 8 (Muthén & Muthén, 2017).” If this is the case you should better explain the reason, please.”

As we use multilevel modeling, individual observations are clustered within countries and all countries automatically receive the same weight. We did not intend to weight by population size, because this would give more weight to larger countries, which is important to calculate population statistics but would not answer the purpose of demonstrating the variation in the effects of voluntary participation across European countries. Our remark referred specifically to our inability to correct for individual-level sampling bias, for instance, that certain population segments in certain countries were more likely to be sampled. No type of weighting is available with multilevel Bayesian estimation in Mplus, which was our method of choice for other reasons (less bias with fewer clusters, fewer convergence problems). 

Nevertheless, we conducted supplementary analyses for Study 1 using ML estimation with individual-level post-stratification weights that aim to reduce sampling errors and non-response bias within each country. As we expected, these models showed massive convergence problems, because ML estimation is not suited for a multilevel analysis with that few units at the between level. Nevertheless, estimates from these models (i.e., the effects of nonpolitical volunteering and political participation, age groups, and their interactions on various well-being outcomes) were very close to the estimates obtained from Bayesian estimation. In most cases, statistical significance (or absence thereof) was also preserved. Apart from convergence problems, weighted analyses resulted in much larger standard errors and therefore less precise estimates in comparison to Bayesian estimation without weights. We mention these supplementary analyses briefly on pp. 17-18.

“Page 15: For missing values you need imputation. You should better explain the following: “Missing values on both dependent and independent variables at 9 the individual level were estimated directly in each model under MAR assumption”

We use full information estimation directly in the model, which is a more economical method than multiple imputation and superior to all other available techniques, such as mean substitution, listwise deletion or single imputation with plausible values (some of these approaches inflate the effective sample size). In the revised manuscript (p. 18), we write: 

“Missing values on both dependent and independent variables at the individual level were estimated directly in each model under MAR assumption: Bayesian estimator is a full information estimator, which uses all available information from each case and implies probable values in the place of missing values (assuming missing values are a function of available covariates) during the estimation of model parameters without inflating the effective sample size (Asparouhov & Muthén, 2010; Enders, 2001).”

“Page 16: “Multiple Regression Results”. Are you sure that you can talk about effects? Don’t you have just correlation?”

We use the word “effects” in a purely technical way to refer to regression coefficients which are not equivalent to bivariate correlations. Besides, we use the terms “associations” and “effects” interchangeably to underscore the cross-sectional nature of our findings. We also acknowledge under Limitations that a cross-sectional design makes causal or even directional inferences impossible. In the revised manuscript, we replaced the word “effect(s)” by “association(s)” wherever this was stylistically and substantially possible (e.g., not where we used established terms such as “main effects” or “interaction effects”; and not where outcomes were omitted, because, in our understanding, the word “effect” allows for the outcomes – effect on what? - to be implied, whereas the word “association” – association with what? - does not).

REVIEWER 2

We thank the reviewer for their overall positive feedback and useful comments.

“what about culture? say

https://en.wikipedia.org/wiki/Inglehart%E2%80%93Welzel_cultural_map_of_the_world

https://en.wikipedia.org/wiki/Hofstede%27s_cultural_dimensions_theory

... and then you enumerate stuff, good, but again what about culture!”

Thank you for bringing in the issue of culture. We screened our cross-national distributions of the effects of volunteering and political participation on various well-being outcomes to check whether they might map onto various cultural dimensions you cited, such as secular-traditional or survival-self-expression (Inglehart) or the six dimensions suggested by Hofstede. We could not recognize any systematic pattern. We refrained from introducing the cultural dimensions as additional country-level variables into our analyses, because this would mean a vast number of post-hoc analyses. Besides, this could be a separate topic for a separate paper. In the revised manuscript, we wrote in the beginning of Discussion (p. 51):

“Just as many European nations appear to be relatively homogenous and distinguishable in terms of central values they hold (Minkov & Hofstede, 2014), they may have developed distinct participation cultures that account for the unique patterns of associations between voluntary participation and well-being that we found in the present studies.”

Furthermore, we added to Future Directions (p. 56):

“The idiosyncrasies of cultures, such as cultural values, might also play a role, although the distributions of the effects of voluntary participation across the ESS countries did not appear to map onto particular dimensions such as traditional versus secular values or power distance (Hofstede Insights, 2022; World Values Survey 7, 2022).”

"Discrepant findings from different countries may result from random sampling

errors"--may but unlikely! any particular evidence of that?"

Well, given that sampling errors would be random, it is difficult to bring evidence for them. This would require conducting a different, more representative cross-national survey with the same measures.

"Plagnol and Huppert (2010) found that formal volunteering was more

10

1 strongly associated with higher SWB, sense of accomplishment, and feelings of doing

2 something worthwhile in European countries with low (vs. high) volunteering rates."

--seems like an important study that you may follow up on and update”

Indeed, we drew on this study and cited it several times in the manuscript. Moreover, taking into account their findings, we conducted supplementary analyses in Study 2 (p. 42):

“We checked whether the moderating effects of GDP could in fact be attributed to lower volunteering rates in low-GDP countries (Hansen et al., 2018; Plagnol & Huppert, 2010). This was indeed the case; however, volunteering rates might mediate the effects of economic development, therefore it would be premature to discard the latter as an explanation for the variation in the effects of nonpolitical volunteering.”

“my only major criticism is that it's too long! see if you can move say 20-80perc of it into appendix/supplementary material

alternatively this could be 2 papers--one on age, one on cross-country differences”

Our manuscript is a classical multi-study paper, whereby each next study builds on the previous one. That is, both Study 2 (cross-national differences in the main effects of volunteering and political participation) and Study 3 (cross-national differences in the age differences in these effects) build on Study 1, which shows that (a) contrary to a widespread opinion, there were few age differences in the effects of volunteering and political participation, and (b) there was a large cross-national variation in both main effects and age differences. Study 2 then investigates country-level factors that might explain the variation in the main effects, whereas Study 3 investigates different country-level factors that might explain the variation in age differences. Study 3 additionally uses some of the country-level variables from Study 2 as control variables or as additional predictors for supplementary analyses (GDP per capita). Splitting the studies in one way or another would, first, constitute a piecemeal publication, and second, effectively increase the overall volume of the manuscripts, because more information from the studies published separately (especially Study 1) would have to flow into the description of the other studies (especially Studies 2 and 3).

As far as we understand, Plos ONE is committed to publish preregistered research, especially with null or controversial findings, and imposes no limitations on the length of submitted manuscripts. One of publication criteria in Plos ONE is “Experiments, statistics, and other analyses are performed to a high technical standard and are described in sufficient detail”. We went through our manuscript and see no room for removing parts of study descriptions, such as Analytical Approach or Results, into supplementary information, because the descriptions in the manuscript are essential for understanding of our methodological approach and findings. (Besides, Plos ONE is an online-only publication, which makes a distinction between the main article and supplementary information even fuzzier.) There are fresh examples of multi-study papers in psychology published in Plos ONE, which have comparable length to our manuscript:

https://journals.plos.org/plosone/article?id=10.1371/journal.pone.0270225

https://journals.plos.org/plosone/article?id=10.1371/journal.pone.0268713

https://journals.plos.org/plosone/article?id=10.1371/journal.pone.0261251

At any rate, in the revised version, the word count was reduced by converting in-text references into numbers, as required by Plos style.

---

## [Editor Report · Decision Letter 1]

23 Jan 2023

Volunteering and political participation are differentially associated with eudaimonic and social well-being across age groups and European countries

PONE-D-22-20390R1

Dear Maria Pavlova,

We’re pleased to inform you that your manuscript has been judged scientifically suitable for publication and will be formally accepted for publication once it meets all outstanding technical requirements.

Kind regards,

Maximo Rossi, PhD Economics

Academic Editor

PLOS ONE

Additional Editor Comments (optional):

Dear Pavlova, thank you for considering the observations that have been made on the paper. There is a reasonable response to each of the suggestions made. Best. Maximo
---

## [Editor Report · Acceptance letter]

25 Jan 2023

PONE-D-22-20390R1 

Volunteering and political participation are differentially associated with eudaimonic and social well-being across age groups and European countries 

Dear Dr. Pavlova:

I'm pleased to inform you that your manuscript has been deemed suitable for publication in PLOS ONE. Congratulations! Your manuscript is now with our production department. 

Kind regards, 

on behalf of

Dr. Maximo Rossi 

Academic Editor

PLOS ONE